# Propagation of Chaos for Mean-Field Langevin Dynamics and its Application to Model Ensemble

Atsushi Nitanda [1 2 3]  Anzelle Lee [4 1 2]  Damian Tan [3 1 2]  Mizuki Sakaguchi [5]  Taiji Suzuki [6 7]

## Abstract

Mean-field Langevin dynamics (MFLD) is an optimization method derived by taking the mean-field limit of noisy gradient descent for two-layer neural networks in the mean-field regime. Recently, the propagation of chaos (PoC) for MFLD has gained attention as it provides a quantitative characterization of the optimization complexity in terms of the number of particles and iterations. A remarkable progress by Chen et al. (2022) showed that the approximation error due to finite particles remains uniform in time and diminishes as the number of particles increases. In this paper, by refining the defective log-Sobolev inequality—a key result from that earlier work—under the neural network training setting, we establish an improved PoC result for MFLD, which removes the exponential dependence on the regularization coefficient from the particle approximation term of the optimization complexity. As an application, we propose a PoC-based model ensemble strategy with theoretical guarantees.

## 1 Introduction

A two layer *mean-field neural network* (MFNN) with $N$ neurons is defined as an empirical average of $N$ functions: $\mathbb{E}_{X \sim \rho_{\mathbf{x}}}[h(X, \cdot)] = \frac{1}{N} \sum_{i=1}^{N} h(x^i, \cdot)$, where each $h(x^i, \cdot)$ represents a single neuron with parameter $x^i$ and $\rho_{\mathbf{x}} = \frac{1}{N} \sum_{i=1}^{N} \delta_{x_i}$ is an empirical distribution. As the number of neurons get infinitely large ($N \to \infty$), the *mean-field limit* is attained: $\rho_{\mathbf{x}} \to \mu$, leading to MFNN having an infinite number of particles: $\mathbb{E}_{X \sim \mu}[h(X, \cdot)]$. Since a distribution $\mu$ parameterizes the model in this mean-field limit, training can now be formulated as the optimization over the space of probability distributions (Nitanda & Suzuki, 2017). Gradient descent for MFNNs exhibits global convergence (Chizat & Bach, 2018; Mei et al., 2018) and adaptivity (Yang & Hu, 2020; Ba et al., 2022). To improve stability during training, one may consider *noisy* gradient training by adding Gaussian noise, giving rise to *mean-field Langevin dynamics* (MFLD) (Mei et al., 2018; Hu et al., 2019). MFLD, with $N = \infty$, also achieves global convergence to the optimal solution (Hu et al., 2019; Jabir et al., 2019), with an exponential convergence rate under the *uniform log-Sobolev inequality* (LSI) (Nitanda et al., 2022; Chizat, 2022) in the continuous-time setting.

However, the mean-field limit attained at $N = \infty$ cannot be accurately replicated in real-life scenarios. When employing a finite-particle system $\rho_{\mathbf{x}}$, the approximation error that arises has been studied in the literature on *propagation of chaos* (PoC) (Sznitman, 1991). In the context of MFLD, Chen et al. (2022); Suzuki et al. (2023a) proved the *uniform-in-time* PoC for the trajectory of MFLD. In particular, in the long-time limit, they established the bounds $\mathcal{L}^{(N)}(\mu_*^{(N)}) - \mathcal{L}(\mu_*) = O\left(\frac{\lambda}{\alpha N}\right)$, where $\alpha \gtrsim \exp\left(-\Theta\left(\frac{1}{\lambda}\right)\right)$ is the LSI constant on *proximal Gibbs distributions*, $\lambda$ is the regularization coefficient, and $\mathcal{L}^{(N)}(\mu_*^{(N)})$ and $\mathcal{L}(\mu_*)$ are the optimal values in finite- and infinite-particle systems. Subsequently, Nitanda (2024) improved upon this result by removing $\alpha$ from the above bound, resulting in $O\left(\frac{1}{N}\right)$. This refinement of the bound is significant as previously, the LSI constant could become exponentially small as $\lambda \to 0$. While Nitanda (2024) also established PoC for the MFLD trajectory by incorporating the *uniform-in-N* LSI (Chewi et al., 2024): $\mathcal{L}^{(N)}(\mu_t^{(N)}) \to \mathcal{L}^{(N)}(\mu_*^{(N)})$, this approach is indirect for showing convergence to the mean-field limit $\mathcal{L}(\mu_*)$ and results in a slower convergence rate over time.

In this work, we further aim to improve PoC for MFLD by demonstrating a faster convergence rate in time, while maintaining the final approximation error $O\left(\frac{1}{N}\right)$ attained at $t = \infty$. We then utilize our result to propose a PoC-based ensemble technique by demonstrating how finite particle systems can converge towards the mean-field limit when merging MFNNs trained in parallel.

---

[1]Institute of High Performance Computing (IHPC), Agency for Science, Technology and Research (A⋆STAR), Singapore [2]Centre for Frontier AI Research (CFAR), Agency for Science, Technology and Research (A⋆STAR), Singapore [3]College of Computing and Data Science, Nanyang Technological University, Singapore [4]School of Computing, National University of Singapore, Singapore [5]Department of Artificial Intelligence, Kyushu Institute of Technology, Japan [6]Department of Mathematical Informatics, The University of Tokyo, Japan [7]Center for Advanced Intelligence Project, RIKEN, Japan. Correspondence to: Atsushi Nitanda <atsushi_nitanda@cfar.a-star.edu.sg>.

*Proceedings of the 42ⁿᵈ International Conference on Machine Learning*, Vancouver, Canada. PMLR 267, 2025. Copyright 2025 by the author(s).

## 1.1 Contributions

The PoC for MFLD (Chen et al., 2022; Suzuki et al., 2023a) consists of particle approximation error $O\left(\frac{\lambda}{\alpha N}\right)$ due to finite-$N$-particles and optimization error $\exp(-\Theta(\lambda\alpha t))$. This result basically builds upon the defective LSI: $\exists\delta > 0$,

$$\frac{1}{N}\mathcal{L}^{(N)}(\mu^{(N)}) - \mathcal{L}(\mu_*) \leq \frac{\delta}{N} + \frac{\lambda}{2\alpha N}\mathrm{FI}(\mu^{(N)}\|\mu_*^{(N)})$$

implicitly established by Chen et al. (2022) under the uniform LSI condition (Nitanda et al., 2022; Chizat, 2022), where FI is Fisher information. The dependence on LSI-constant $\alpha$ in $O\left(\frac{\lambda}{\alpha N}\right)$ of PoC is basically inherited from $\delta$. In our work, we first remove the dependence on $\alpha$ from $\delta$ by introducing *uniform directional LSI* (Assumption 3.2) in training MFNNs setting. Based on this defective LSI, we then derive an improved PoC for MFLD where the particle approximation error is $O\left(\frac{1}{N}\right)$. Similar to Nitanda (2024), this improvement exponentially reduces the required number of particles since the constant $\alpha \gtrsim \exp\left(-\Theta(\frac{1}{\lambda})\right)$ can exponentially decrease as $\lambda \to \infty$. Moreover, our result demonstrates a faster optimization speed compared to Nitanda (2024); Chewi et al. (2024) due to a different exponent $\alpha$ in the optimization error terms: $\exp(-\Theta(\lambda\alpha t))$. In our analysis, $\alpha$ is a constant of the uniform directional LSI, which is larger than the LSI constant on $\mu_*^{(N)}$ appearing in the optimization error in Nitanda (2024); Chewi et al. (2024) (see the discussion following Theorem 3.8).

Next, we translate the PoC result regarding objective gap into the point-wise and uniform model approximation errors: $|\mathbb{E}_{X\sim\rho_{\mathbf{x}}}[h(X,z)] - \mathbb{E}_{X\sim\mu_*}[h(X,z)]|$ and $\|\mathbb{E}_{X\sim\rho_{\mathbf{x}}}[h(X,\cdot)] - \mathbb{E}_{X\sim\mu_*}[h(X,\cdot)]\|_\infty$ useful for obtaining generalization error bound on classification task (Suzuki et al., 2023b; Nitanda et al., 2024). Again, the bound consists of the sum of particle approximation and optimization error terms. Compared to the previous results (Suzuki et al., 2023a;b), our bound is tighter since the particle approximation term is independent of the LSI-constant. This improvement directly eliminates the requirement for an exponential number of neurons with respect to dimension $d$ in their learning setup (e.g., $k$-parity problems (Suzuki et al., 2023b)). We also propose a PoC-based model ensemble method to further reduce the model approximation error and empirically verify its performance on synthetic datasets. To our knowledge, our study is the first to provide a theoretical guarantee for model ensembling of MFNNs using PoC results. Going beyond the scope of the theory, we examine the applicability of our method to merging LoRA parameters for language models.

- We demonstrate an improved PoC for MFLD (Theorem 3.8) under uniform directional LSI condition (Assumption 3.2). This improvement removes the dependence on LSI constant $\alpha \gtrsim \exp\left(-\Theta\left(\frac{1}{\lambda}\right)\right)$ from the parti-

cle approximation error in Chen et al. (2022); Suzuki et al. (2023a) and accelerates the optimization speed in Nitanda (2024); Chewi et al. (2024).

- We translate the PoC result regarding objective gap into point-wise and uniform model approximation errors (Theorems 4.3, 4.4, and 4.7). These results also remove the dependence on the LSI constant from the particle approximation terms in the previous model approximation errors (Suzuki et al., 2023a;b).

- We propose an ensembling method for MFNNs trained in parallel to reduce approximation error, providing theoretical guarantees (Theorem 4.4, 4.7) and empirical verification on synthetic datasets. Moreover, going beyond the theoretical framework, we apply our method to merge multiple LoRA parameters of language models and observe improved prediction performance.

## 1.2 Notations

We use lowercase letters such as $x$ for vectors and uppercase letters such as $X$ for random variables $\mathbb{R}^d$, respectively. The boldface is used for tuples of them like $\mathbf{x} = (x^1, \ldots, x^N) \in \mathbb{R}^{Nd}$ and $\mathbf{X} = (X^1, \ldots, X^N)$. Given $\mathbf{x} = (x^i)_{i=1}^N$, $\mathbf{x}^{-i}$ denotes $(x^1, \ldots, x^{i-1}, x^{i+1}, \ldots, x^N)$. $\|\cdot\|_2$ denotes the Euclidean norm. $\mathcal{P}_2(\mathbb{R}^d)$ denotes the set of probability distributions with finite second moment on $\mathbb{R}^d$. For probability distributions $\mu, \nu \in \mathcal{P}_2(\mathbb{R}^d)$, we define Kullback-Leibler (KL) divergence (a.k.a. relative entropy) by $\mathrm{KL}(\mu\|\nu) = \int \mathrm{d}\mu(x) \log\frac{\mathrm{d}\mu}{\mathrm{d}\nu}(x)$ and define Fisher information by $\mathrm{FI}(\mu\|\nu) = \int \mathrm{d}\mu(x)\|\nabla\log\frac{\mathrm{d}\mu}{\mathrm{d}\nu}(x)\|_2^2$. Ent denotes the negative entropy: $\mathrm{Ent}(\mu) = \int \mu(\mathrm{d}x)\log\frac{\mathrm{d}\mu}{\mathrm{d}x}(x)$. We denote $\langle f, m\rangle = \int f(x)m(\mathrm{d}x)$ for a (signed) measure $m$ and integrable function $f$ on $\mathbb{R}^d$. Given $\mathbf{x} = (x^1, \ldots, x^N) \in \mathbb{R}^{Nd}$, we write an empirical distribution supported on $\mathbf{x}$ as $\rho_{\mathbf{x}} = \frac{1}{N}\sum_{i=1}^N \delta_{x^i}$.

## 2 Preliminaries

In this section, we explain the problem setting and give a brief literature review of MFLD and PoC. See Appendix C for additional background information.

## 2.1 Problem setting

For a functional $G : \mathcal{P}_2(\mathbb{R}^d) \to \mathbb{R}$, we say $G$ is differentiable when there exists a functional (referred to as a *first variation*): $\frac{\delta G}{\delta\mu} : \mathcal{P}_2(\mathbb{R}^d) \times \mathbb{R}^d \ni (\mu, x) \mapsto \frac{\delta G(\mu)}{\delta\mu}(x) \in \mathbb{R}$ such that for any $\mu, \mu' \in \mathcal{P}_2(\mathbb{R}^d)$,

$$\frac{\mathrm{d}G(\mu + \epsilon(\mu' - \mu))}{\mathrm{d}\epsilon}\bigg|_{\epsilon=0} = \int \frac{\delta G(\mu)}{\delta\mu}(x)(\mu' - \mu)(\mathrm{d}x),$$

and say $G$ is linearly convex when for any $\mu, \mu' \in \mathcal{P}_2(\mathbb{R}^d)$,

$$G(\mu') \geq G(\mu) + \int \frac{\delta G(\mu)}{\delta \mu}(x)(\mu' - \mu)(\mathrm{d}x). \quad (1)$$

Given a differentiable and linearly convex functional $F_0 : \mathcal{P}_2(\mathbb{R}^d) \to \mathbb{R}$ and $\lambda > 0$, we consider the minimization problem of an entropy-regularized convex functional:

$$\min_{\mu \in \mathcal{P}_2(\mathbb{R}^d)} \left\{ \mathcal{L}(\mu) = F_0(\mu) + \mathbb{E}_{X \sim \mu}[r(X)] + \lambda \mathrm{Ent}(\mu) \right\}, \quad (2)$$

where $r : \mathbb{R}^d \to \mathbb{R}$ is a $\lambda'$-strongly convex function (e.g., $r(x) = \lambda' \|x\|_2^2$ ($\lambda' > 0$)). We set $F(\mu) = F_0(\mu) + \mathbb{E}_\mu[r(X)]$. A typical example of $F_0$ is an empirical risk of the two-layer mean-field neural network (see Example 3.5). Throughout the paper, we assume that the solution $\mu_* \in \mathcal{P}_2(\mathbb{R}^d)$ of the problem (2) exists and make the following regularity assumption (Chizat, 2022; Nitanda et al., 2022; Chen et al., 2023) under which $\mu_*$ is unique and satisfies the optimality condition: $\mu_* \propto \exp\left(-\frac{1}{\lambda}\frac{\delta F(\mu_*)}{\delta \mu}\right)$ (see Hu et al. (2019); Chizat (2022) for the details).

**Assumption 2.1.** There exists $C_1, C_2 > 0$ such that for any $\mu \in \mathcal{P}_2(\mathbb{R}^d)$, $x \in \mathbb{R}^d$, $\left|\nabla \frac{\delta F_0(\mu)}{\delta \mu}(x)\right| \leq C_1$ and for any $\mu, \mu' \in \mathcal{P}_2(\mathbb{R}^d)$, $x, x' \in \mathbb{R}^d$,

$$\left\|\nabla \frac{\delta F_0(\mu)}{\delta \mu}(x) - \nabla \frac{\delta F_0(\mu')}{\delta \mu}(x')\right\|_2$$
$$\leq C_2 \left(W_2(\mu, \mu') + \|x - x'\|_2\right),$$

where $W_2$ is the 2-Wasserstein distance.

## 2.2 Mean-field Langevin dynamics and uniform-in-time propagation of chaos

First, consider the finite-particle setting $\rho_{\mathbf{x}} = \frac{1}{N}\sum_{i=1}^N \delta_{x^i}$ for $\mathbf{x} = (x^i)_{i=1}^N \in \mathbb{R}^{dN}$ and the following noisy gradient descent for $F(\rho_{\mathbf{x}})$. Given $k$-th iteration $\mathbf{X}_k = (X_k^1, \ldots, X_k^N)$, for each $i \in \{1, 2, \ldots, N\}$, we perform

$$X_{k+1}^i = X_k^i - \eta \nabla \frac{\delta F(\rho_{\mathbf{X}_k})}{\delta \mu}(X_k^i) + \sqrt{2\lambda\eta}\xi_k^i, \quad (3)$$

where $\xi_k^i \sim \mathcal{N}(0, I_d)$ ($i \in \{1, 2, \ldots, N\}$) are i.i.d. standard normal random variables and the gradient in the RHS is taken for the function: $\frac{\delta F(\rho_{\mathbf{X}_t})}{\delta \mu}(\cdot) : \mathbb{R}^d \to \mathbb{R}$. The continuous-time representation of Eq. (3) is given by the $N$-tuple of SDEs $\{\mathbf{X}_t\}_{t\geq 0} = \{(X_t^1, \ldots, X_t^N)\}_{t\geq 0}$:

$$\mathrm{d}X_t^i = -\nabla \frac{\delta F(\rho_{\mathbf{X}_t})}{\delta \mu}(X_t^i)\mathrm{d}t + \sqrt{2\lambda}\mathrm{d}W_t^i, \quad (4)$$

where $\{W_t^i\}_{t\geq 0}$, ($i \in \{1, \ldots, N\}$) are independent standard Brownian motions. Note that Eq. (4) is equivalent to the Langevin dynamics $\mathrm{d}\mathbf{X}_t = -N\nabla_{\mathbf{X}}F(\rho_{\mathbf{X}_t})\mathrm{d}t +$

$\sqrt{2\lambda}\mathrm{d}\mathbf{W}_t$ on $\mathbb{R}^{dN}$, where $\{\mathbf{W}_t\}_{t\geq 0}$ is the standard Brownian motion on $\mathbb{R}^{dN}$ since $N\nabla_{x^i}F(\rho_{\mathbf{x}}) = \nabla \frac{\delta F(\mu_{\mathbf{x}})}{\delta \mu}(x^i)$ (Chizat, 2022). Therefore, $\mu_t^{(N)} = \mathrm{Law}(\mathbf{X}_t)$ converges to the Gibbs distribution

$$\frac{\mathrm{d}\mu_*^{(N)}}{\mathrm{d}\mathbf{x}}(\mathbf{x}) \propto \exp\left(-\frac{N}{\lambda}F(\rho_{\mathbf{x}})\right).$$

which minimizes the following entropy-regularized linear functional defined on $\mathcal{P}_2(\mathbb{R}^{dN})$: for $\mu^{(N)} \in \mathcal{P}_2(\mathbb{R}^{dN})$,

$$\mathcal{L}^{(N)}(\mu^{(N)}) = N\mathbb{E}_{\mathbf{X} \sim \mu^{(N)}}[F(\rho_{\mathbf{X}})] + \lambda \mathrm{Ent}(\mu^{(N)}). \quad (5)$$

Next, we take the mean-field limit: $N \to \infty$ under which Eq. (4) converges to the MFLD that solves the problem Eq. (2);

$$\mathrm{d}X_t = -\nabla \frac{\delta F}{\delta \mu}(\mu_t)(X_t)\mathrm{d}t + \sqrt{2\lambda}\mathrm{d}W_t, \quad \mu_t = \mathrm{Law}(X_t), \quad (6)$$

where $\{W_t\}_{t\geq 0}$ is the $d$-dimensional standard Brownian motion with $W_0 = 0$. Under the log-Sobolev inequality on the proximal Gibbs distribution $\hat{\mu} \propto \exp\left(-\frac{1}{\lambda}\frac{\delta F(\mu)}{\delta \mu}\right)$, Nitanda et al. (2022); Chizat (2022) showed the exponential convergence of the objective gap $\mathcal{L}(\mu_t) - \mathcal{L}(\mu_*)$, where $\mu_* = \mathrm{argmin}_{\mu \in \mathcal{P}_2(\mathbb{R}^d)} \mathcal{L}(\mu)$.

Therefore, $\frac{1}{N}\mathcal{L}^{(N)}(\mu_k^{(N)})$, where $\mu_k^{(N)} = \mathrm{Law}(\mathbf{X}_k)$, is expected to approximate $\mathcal{L}(\mu_*)$ through the time and mean-field limit $k \to \infty, N \to \infty$, leading to the natual question:

*What is the convergence rate of $\frac{1}{N}\mathcal{L}^{(N)}(\mu_k^{(N)})$ to $\mathcal{L}(\mu_*)$?*

This approximation error has been studied in the literature of PoC. Recently, Suzuki et al. (2023a) proved the following uniform-in-time PoC for Eq. (3) by using the techniques in Chen et al. (2022):

$$\frac{1}{N}\mathcal{L}^{(N)}(\mu_k^{(N)}) - \mathcal{L}(\mu_*) \leq \exp\left(-\lambda\alpha\eta k/2\right)\Delta_0^{(N)} + \delta_{\eta,N}, \quad (7)$$

where $\Delta_0^{(N)} = \frac{1}{N}\mathcal{L}^{(N)}(\mu_0^{(N)}) - \mathcal{L}(\mu_*)$ is the initial gap and $\delta_{\eta,N} = \frac{(\lambda\eta + \eta^2)D_1}{\lambda\alpha} + \frac{\lambda D_2}{\alpha N}$ ($\exists D_1, D_2 > 0$) is the discretization error in time and space. The continuous-time counterpart ($\eta \to 0$) was proved by Chen et al. (2022). The typical estimation of LSI-constant $\alpha \gtrsim \exp(-\Theta(1/\lambda))$ (e.g., Theorem 1 in Suzuki et al. (2023a)) using Holley and Stroock argument (Holley & Stroock, 1987) or Miclo's trick (Bardet et al., 2018)) suggests the exponential blow-up of the particle approximation error $\frac{\lambda D_2}{\alpha N}$ in Eq. (7) as $\lambda \to 0$.

Afterward, this exponential dependence was removed by Nitanda (2024); Chewi et al. (2024) that evaluate the particle approximation error at the solution: $\frac{1}{N}\mathcal{L}^{(N)}(\mu_*^{(N)}) - \mathcal{L}(\mu_*)$

and optimization error: $\frac{1}{N}\left(\mathcal{L}^{(N)}(\mu_k^{(N)}) - \mathcal{L}^{(N)}(\mu_*^{(N)})\right)$, respectively. In the risk minimization problem setting, Nitanda (2024) proved $\frac{1}{N}\mathcal{L}^{(N)}(\mu_*^{(N)}) - \mathcal{L}(\mu_*) \leq \frac{C}{N}$ ($\exists C > 0$) and Chewi et al. (2024) proved uniform-in-$N$ LSI on $\mu_*^{(N)} \in \mathbb{R}^{dN}$ with the constant estimation $\bar{\alpha} \gtrsim \frac{\lambda'}{\lambda}\exp\left(-O\left(\frac{1}{\lambda'} + \frac{1}{\lambda\lambda'} + \frac{1}{\lambda^2\lambda'^3}\right)\right)$, leading to the $N$-independent convergence rate of $\frac{1}{N}\mathcal{L}^{(N)}(\mu_k^{(N)}) - \mathcal{L}(\mu_*)$ up to the particle approximation error $C/N$ plus time-discretization error.

## 3  Main Result I: Improved Propagation of Chaos for Mean-field Neural Network

In this section, we present an improved propagation-of-chaos for the mean-field Langevin dynamics under the uniform directional LSI introduced below.

**Definition 3.1.** For $\mathbf{x}^{-i} = (x^1, \ldots, x^{i-1}, x^{i+1}, \ldots, x^N)$ ($i \in \{1, 2, \ldots, N\}$), we define a *conditional Gibbs distribution* $\nu_{i|-i}(\cdot|\mathbf{x}^{-i})$ on $\mathbb{R}^d$ by

$$\frac{\mathrm{d}\nu_{i|-i}}{\mathrm{d}x}(x|\mathbf{x}^{-i}) = \frac{\exp\left(-\frac{N}{\lambda}F(\rho_{x\cup\mathbf{x}^{-i}})\right)}{\int \exp\left(-\frac{N}{\lambda}F(\rho_{\tilde{x}\cup\mathbf{x}^{-i}})\right)\mathrm{d}\tilde{x}},$$

where $\rho_{x\cup\mathbf{x}^{-i}} = \frac{1}{N}\sum_{j\neq i}\delta_{x^j} + \frac{1}{N}\delta_x$.

**Assumption 3.2** (Uniform directional LSI)**.** There exists a constant $\alpha > 0$ such that for any $\mathbf{x} \in \mathbb{R}^{dN}$ and $i \in \{1, 2, \ldots, N\}$, $\nu_{i|-i}(\cdot|\mathbf{x}^{-i})$ satisfies the LSI with the constant $\alpha$; for all $\mu \in \mathcal{P}_2(\mathbb{R}^d)$ absolutely continuous w.r.t. $\nu_{i|-i}(\cdot \mid \mathbf{x}^{-i})$, it follows that

$$\mathrm{KL}(\mu\|\nu_{i|-i}(\cdot|\mathbf{x}^{-i})) \leq \frac{1}{2\alpha}\mathrm{FI}(\mu\|\nu_{i|-i}(\cdot|\mathbf{x}^{-i})).$$

**Remark.** Wang (2024) also introduced the conditional Gibbs distribution and imposed a Poincaré inequality on it.

We also make the following assumptions.

**Assumption 3.3.** A functional $F_0(\mu)$ is differentiable and linearly convex.

The nonlinearity of $F_0$ is the key to the PoC analysis for mean-field models, thereby motivating the use of the Bregman divergence associated with $F_0$ (Nitanda, 2024); for distributions $\mu, \mu' \in \mathcal{P}_2(\mathbb{R}^d)$,

$$B_{F_0}(\mu, \mu') = F_0(\mu) - F_0(\mu') - \left\langle\frac{\delta F_0(\mu')}{\delta\mu}, \mu - \mu'\right\rangle.$$

**Assumption 3.4.** There exists a constant $B > 0$ such that for any $\mathbf{x} \in \mathbb{R}^{dN}$, $x \in \mathbb{R}^d$, and $i \in \{1, 2, \ldots, N\}$,

$$B_{F_0}(\rho_{x\cup\mathbf{x}^{-i}}, \rho_{\mathbf{x}}) \leq \frac{B}{N^2}.$$

Here, we give a connection between a conditional Gibbs distribution $\nu_{i|-i}(\cdot|\mathbf{x}^{-i})$ and proximal Gibbs distribution $\hat{\rho}_{\mathbf{x}}$ using the Bregman divergence. Given $\mathbf{x} = (x^1, \ldots, x^N)$, the following relationship holds as a probability distribution over $x \in \mathbb{R}$:

$$\begin{aligned}
\frac{\mathrm{d}\nu_{i|-i}}{\mathrm{d}x}(x|\mathbf{x}^{-i}) &\propto \exp\left(-\frac{N}{\lambda}F(\rho_{x\cup\mathbf{x}^{-i}})\right) \\
&= \exp\left(-\frac{N}{\lambda}\left(F(\rho_{\mathbf{x}}) + \left\langle\frac{\delta F(\rho_{\mathbf{x}})}{\delta\mu}, \rho_{x\cup\mathbf{x}^{-i}} - \rho_{\mathbf{x}}\right\rangle\right.\right. \\
&\qquad\left.\left. + B_F(\rho_{x\cup\mathbf{x}^{-i}}, \rho_{\mathbf{x}})\right)\right) \\
&\propto \exp\left(-\frac{1}{\lambda}\frac{\delta F(\rho_{\mathbf{x}})}{\delta\mu}(x) - \frac{N}{\lambda}B_F(\rho_{x\cup\mathbf{x}^{-i}}, \rho_{\mathbf{x}})\right) \\
&\propto \frac{\mathrm{d}\hat{\rho}_{\mathbf{x}}}{\mathrm{d}x}(x)\exp\left(-\frac{N}{\lambda}B_F(\rho_{x\cup\mathbf{x}^{-i}}, \rho_{\mathbf{x}})\right). \qquad (8)
\end{aligned}$$

This connection is useful for deriving the LSI on $\nu_{i|-i}$ from the uniform LSI on the proximal Gibbs distributions $\hat{\rho}_{\mathbf{x}}$ (see Nitanda et al. (2022)), in combination with Assumption 3.4 and Holley-Strook argument.

We give an example of training MFNNs that satisfies Assumptions 3.2, 3.3, and 3.4.

**Example 3.5** (Training MFNN)**.** Let $\mathcal{Y} \subset \mathbb{R}$ be a label space, $\mathcal{Z} \subset \mathbb{R}^{d'}$ be an input data space, $h(x, \cdot) : \mathcal{Z} \to \mathbb{R}$ be a function parameterized by $x \in \mathbb{R}^d$, and $\ell(\cdot, \cdot) : \mathbb{R} \times \mathbb{R} \to \mathbb{R}$ is a loss function. Given training examples $\{(z_j, y_j)\}_{j=1}^n \subset \mathcal{Z} \times \mathcal{Y}$, we consider the empirical risk:

$$F_0(\mu) = \frac{1}{n}\sum_{j=1}^n \ell\left(\mathbb{E}_{X\sim\mu}[h(X, z_j)], y_j\right),$$

and $L_2$-regularizaton $r(x) = \lambda'\|x\|_2^2$. We assume that $\sup_{x\in\mathbb{R}^d, z\in\mathcal{Z}}|h(x, z)| \leq R$ and that for any $y \in \mathbb{R}$, $\ell(\cdot, y)$ is convex and $L$-smooth; there exists $L > 0$ such that for any $a, b \in \mathbb{R}$, $\ell(b, y) \leq \ell(a, y) + \frac{\partial\ell(a, y)}{\partial a}(b - a) + \frac{L}{2}|b - a|^2$. Applying this $L$-smoothness with $a = \mathbb{E}_{\rho_{\mathbf{x}}}[h(X, z_j)], b = \mathbb{E}_{\rho_{x\cup\mathbf{x}^{-i}}}[h(X, z_j)], y = y_j$ and taking average over $j \in \{1, 2, \ldots, n\}$, we get $B_{F_0}(\rho_{x\cup\mathbf{x}^{-i}}, \rho_{\mathbf{x}}) \leq \frac{L}{2n}\sum_{j=1}^n\left|\frac{h(x^i, z_j)}{N}\right|^2 \leq \frac{LR^2}{2N^2}$. Therefore, the Holley-Stroock argument (Holley & Stroock, 1987) with Eq. (8) implies an LSI constant $\alpha = \alpha_0\exp(-\frac{2LR^2}{\lambda N})$ converging to $\alpha_0$ as $N \to \infty$, where $\alpha_0 = \frac{2\lambda'}{\lambda\exp(O(\lambda^{-1}))}$ is the LSI-constant of the proximal Gibbs distribution $\hat{\rho}_{\mathbf{x}}$.

The following defective entropy sandwich and defective LSI are key results in studying MFLD in the finite-particle setting. The proofs can be found in Appendix A.1. For $\mathbf{X} \sim \mu^{(N)}$, we denote by $\mu_{i|-i}^{(N)}(\cdot|\mathbf{x}^{-i})$ the conditional distribution of $X^i$ conditioned by $\mathbf{X}^{-i} = \mathbf{x}^{-i}$.

**Lemma 3.6** (Defective entropy sandwich). *Suppose Assumption 3.4 holds. Then, for any $\mu^{(N)} \in \mathcal{P}_2(\mathbb{R}^{dN})$,*

$$\frac{\lambda}{N} \mathrm{KL}(\mu^{(N)} \| \mu_*^{\otimes N}) + \mathbb{E}_{\mathbf{X} \sim \mu^{(N)}} [B_{F_0}(\rho_{\mathbf{X}}, \mu_*)]$$

$$= \frac{1}{N} \mathcal{L}^{(N)}(\mu^{(N)}) - \mathcal{L}(\mu_*)$$

$$\leq \frac{B}{N} + \frac{\lambda}{N} \sum_{i=1}^{N} \mathbb{E}_{\mathbf{X} \sim \mu^{(N)}} \left[ \mathrm{KL}(\mu_{i|-i}^{(N)}(\cdot | \mathbf{X}^{-i}) \| \nu_{i|-i}(\cdot | \mathbf{X}^{-i})) \right].$$

The lemma 3.6 can be viewed as the finite-particle counterpart of the entropy sandwich established in Nitanda et al. (2022); Chizat (2022):

$$\lambda \mathrm{KL}(\mu \| \mu_*) \leq \mathcal{L}(\mu) - \mathcal{L}(\mu_*) \leq \mathrm{KL}(\mu \| \hat{\mu})$$

Eq. (8) says that the conditional distribution $\nu_{i|-i}(\cdot | \mathbf{X}^{-i})$ approximates the proximal Gibbs distribution $\hat{\rho}_{\mathbf{X}}$, where $\rho_{\mathbf{X}}$ is an empirical distribution consisting of $\mathbf{X}$. Therefore, we expect $\nu_{i|-i}(\cdot | \mathbf{X}^{-i})$ to play a role analogous to the proximal distribution and lead to an entropy sandwich. In fact, we can confirm that Lemma 3.6 with $\mu^{(N)} = \mu^{\otimes N}$ reproduces the above entropy sandwich in the infinite-particle system by taking $N \to \infty$ under regular conditions.

The defective LSI was originally established by Chen et al. (2022). We here derive an improved variant built upon the defective entropy sandwich. The proof can be found in Appendix A.1.

**Lemma 3.7** (Defective LSI). *Suppose Assumptions 3.2, 3.3, and 3.4 hold. Then, it follows that for any $\mu^{(N)} \in \mathcal{P}_2(\mathbb{R}^{dN})$,*

$$\frac{\lambda}{N} \mathrm{KL}(\mu^{(N)} \| \mu_*^{\otimes N}) + \mathbb{E}_{\mathbf{X} \sim \mu^{(N)}} [B_{F_0}(\rho_{\mathbf{X}}, \mu_*)]$$

$$= \frac{1}{N} \mathcal{L}^{(N)}(\mu^{(N)}) - \mathcal{L}(\mu_*) \leq \frac{B}{N} + \frac{\lambda}{2\alpha N} \mathrm{FI}(\mu^{(N)} \| \mu_*^{(N)}).$$

Lemma 3.7 gives an approximation error bound between $\mu^{(N)}$ and $\mu_*^{\otimes N}$, which shrinks up to $B/N$ error as $\mu^{(N)} \to \mu_*^{(N)}$ and shrinks to zero by additionally taking $N \to \infty$, meaning that each particle of the system $(X^1, \ldots, X^N) \sim \mu^{(N)}$ becomes independent to each other. Compared to the original result (Chen et al., 2022), the particle approximation term $B/N$ is independent of $\alpha$, similar to Nitanda (2024). Note that whereas Nitanda (2024) only consider the case of $\mu^{(N)} = \mu_*^{(N)}$, our result allows for any distribution $\mu^{(N)}$ at the cost of the Fisher information $\mathrm{FI}(\mu^{(N)} \| \mu_*^{(N)})$. Lemma 3.7 can be indeed regarded as an extended LSI on the finite-particle system and nonlinear mean-field objective, where Fisher information is lower bounded by the optimality gap up to $B/N$ error. In particular, when $F_0$ is the linear functional: $F_0(\mu) = \mathbb{E}_\mu[f]$ ($\exists f : \mathbb{R}^d \to \mathbb{R}$), the lemma reproduces the standard LSI on $\mu_*^{(N)}$:

$$\mathrm{KL}(\mu^{(N)} \| \mu_*^{(N)}) \leq \frac{1}{2\alpha} \mathrm{FI}(\mu^{(N)} \| \mu_*^{(N)})$$

because $\mu_*^{(N)} = \mu_*^{\otimes N}$, $B_{F_0} = 0$, and $B = 0$ in this case.

Therefore, Lemma 3.7 is instrumental in the computational complexity analysis of MFLD in the finite-particle setting as shown in the following theorem. We set $\Delta_0^{(N)} = \frac{1}{N} \mathcal{L}^{(N)}(\mu_0^{(N)}) - \mathcal{L}(\mu_*)$.

**Theorem 3.8** (Propagation chaos for MFLD). *Suppose Assumptions 2.1, 3.2, 3.3, and 3.4 hold and consider the $L_2$-regularization: $r(x) = \lambda' \|x\|_2^2$ ($\lambda' > 0$). Then,*

1. *MFLD in the continuous-time (4) satisfies*

$$\frac{1}{N} \mathcal{L}^{(N)}(\mu_t^{(N)}) - \mathcal{L}(\mu_*) \leq \frac{B}{N} + \exp(-2\alpha\lambda t)\Delta_0^{(N)}.$$

2. *MFLD in the discrete-time (3) with $\eta\lambda' < 1/2$ satisfies*

$$\frac{1}{N} \mathcal{L}^{(N)}(\mu_k^{(N)}) - \mathcal{L}(\mu_*) \leq \frac{B}{N} + \frac{\delta_\eta}{\alpha\lambda} + \exp(-\alpha\lambda\eta k)\Delta_0^{(N)},$$

*where $\delta_\eta = 8\eta(C_2^2 + \lambda'^2)(\eta C_1^2 + \lambda d) + 32\eta^2\lambda'^2(C_2^2 + \lambda'^2)\left(\frac{\mathbb{E}[\|\mathbf{X}_0\|_2^2]}{N} + \frac{1}{\lambda'}\left(\frac{C_1^2}{4\lambda'} + \lambda d\right)\right)$.*

*Proof.* We here demonstrate the convergence in the continuous-time setting. The distribution $\mu_t^{(N)} = \mathrm{Law}(\mathbf{X}_t)$ of Eq. (4) satisfies the following Fokker-Planck equation:

$$\frac{\partial \mu_t^{(N)}}{\partial t} = \lambda \nabla \cdot \left( \mu_t^{(N)} \log \frac{\mathrm{d}\mu_t^{(N)}}{\mathrm{d}\mu_*^{(N)}} \right).$$

By the standard argument of Langevin dynamics (e.g., Vempala & Wibisono (2019)) and Lemma 3.7, we get

$$\frac{\mathrm{d}}{\mathrm{d}t}(\mathcal{L}^{(N)}(\mu_t^{(N)}) - N\mathcal{L}(\mu_*) - B) = -\lambda^2 \mathrm{FI}(\mu_t^{(N)} \| \mu_*^{(N)})$$

$$\leq -2\alpha\lambda(\mathcal{L}^{(N)}(\mu_t^{(N)}) - N\mathcal{L}(\mu_*) - B).$$

Then, the statement follows from a direct application of the Grönwall's inequality. The convergence in the discrete-time is also proved by incorporating one-step iterpolation argument. See Appendix A.1. $\square$

From this result, we see that MFLD indeed induces the PoC regarding KL-divergence. In fact, the following inequality, which is a direct consequence of Lemma 3.7 with Theorem 3.8 in the continuous-time, shows that the particles $(X_t^i)_{i=1}^N \sim \mu_t^{(N)}$ become independent as $t \to \infty$ and $N \to \infty$:

$$\frac{1}{N} \mathrm{KL}(\mu_t^{(N)} \| \mu_*^{\otimes N}) \leq \frac{B}{\lambda N} + \exp(-2\alpha\lambda t)\frac{\Delta_0^{(N)}}{\lambda}. \quad (9)$$

We note that the particle approximation term $B/N$ in Theorem 3.8 is independent of LSI-constants, whereas the error $O(\frac{\lambda}{\alpha' N})$ obtained in Suzuki et al. (2023a) scales inversely

with an LSI constant $\alpha'$ on the proximal Gibbs distribution which can be exponentially small as $\lambda \to 0$. Whereas the term $B/N$ is comparable to Nitanda (2024); Chewi et al. (2024), our convergence rate $\exp(-2\alpha\lambda t)$ in optimization is faster since their results rely on the LSI constant $\bar{\alpha}$ on $\mu_*^{(N)}$ which is smaller than $\alpha_0$ in Example 3.5[1]. For instance, Chewi et al. (2024) estimated the LSI constant $\bar{\alpha} \gtrsim \frac{\lambda'}{\lambda} \exp\left(-O\left(\frac{1}{\lambda'} + \frac{1}{\lambda\lambda'} + \frac{1}{\lambda^2\lambda'^3}\right)\right)$.

# 4 Main Result II: Model Approximation Error and PoC-based Model Ensemble

In this section, we study how MFNNs trained with MFLD approximate the mean-field limit: $\mathbb{E}_{X\sim\mu_*}[h(X,z)]$. Moreover, we present a PoC-based model ensemble method that further reduces the error. Throughout this section, we focus on training MFNNs (Example 3.5) and suppose $\sup_{x\in\mathbb{R}^d, z\in\mathcal{Z}} |h(x,z)| \leq R$.

## 4.1 Point-wise model approximation error

We consider point-wise model approximation error between $\mathbb{E}_{X\sim\rho_\mathbf{X}}[h(X,z)] = \frac{1}{N}\sum_{i=1}^{N} h(X^i, z)$ and $\mathbb{E}_{X\sim\mu_*}[h(X,z)]$ on each point $z \in \mathcal{Z}$, where $\mathbf{X} = (X^1, \ldots, X^N) \sim \mu^{(N)}$. The error usually consists of the bias and variance terms where the bias means the difference between $\mu^{(N)}$ and $\mu_*^{\otimes N}$ and the variance is due to finite-$N$ particles. In general, it is not straightforward to show the variance reduction as $N \to \infty$ since $X_i$ $(i = 1, 2, \ldots, N)$ are not independent, and hence can exhibit positive correlation. However, in our setting, PoC helps to reduce the correlation among particles, resulting in better approximation error via the variance reduction.

Since we are concerned with the correlation between each pair of particles, we reinterpret $\mathrm{KL}(\mu^{(N)}\|\mu_*^{\otimes N})$ as the gap between their marginal distributions. For each index subset $S \subset \{1, \ldots, N\}$, we denote by $\mu_S^{(N)}$ the marginal distribution of $\mu^{(N)}$ on $S$ and write $\mu_{1:s}^{(N)} = \mu_{\{1,\ldots,s\}}^{(N)}$, $\mu_i^{(N)} = \mu_{\{i\}}^{(N)}$, $\mu_{i,j}^{(N)} = \mu_{\{i,j\}}^{(N)}$ for simplicity. We say the distribution $\mu^{(N)}$ is *exchangeable* if the laws of $(X_{\sigma(1)}, \ldots, X_{\sigma(N)})$ and $(X_1, \ldots, X_N)$ are identical for all permutation $\sigma : \{1, 2, \ldots, N\} \to \{1, 2, \ldots, N\}$.

**Lemma 4.1.** *For any integers $s, N \in \mathbb{N}$ such that $s \leq N$, it follows that*

$$\frac{N}{s\binom{N}{s}} \sum_{|S|=s} \mathrm{KL}(\mu_S^{(N)}\|\mu_*^{\otimes s}) \leq \mathrm{KL}(\mu^{(N)}\|\mu_*^{\otimes N}).$$

[1]However, we note PoC result obtained by Chewi et al. (2024) is applicable to non-bounded activation functions such as ReLU.

*In particular, if $\mu^{(N)}$ is exchangeable, we get*

$$\frac{N}{s}\mathrm{KL}(\mu_{1:s}^{(N)}\|\mu_*^{\otimes s}) \leq \mathrm{KL}(\mu^{(N)}\|\mu_*^{\otimes N}).$$

*Proof.* The assertion holds by the direct application of Han's inequality. See Appendix A.2. □

We here give the model approximation bound using $\mathrm{KL}(\mu^{(N)}\|\mu_*^{\otimes N})$ with the proof to show how the above lemma helps to control the correlation across particles.

**Proposition 4.2.** *Suppose $\mu^{(N)}$ is exchangeable. Then, it follows that for any $z \in \mathcal{Z}$,*

$$\mathbb{E}_{\mathbf{X}\sim\mu^{(N)}}\left[\left(\mathbb{E}_{X\sim\rho_\mathbf{X}}[h(X,z)] - \mathbb{E}_{X\sim\mu_*}[h(X,z)]\right)^2\right]$$

$$\leq \frac{4R^2}{N} + 8R^2\sqrt{\frac{\mathrm{KL}(\mu^{(N)}\|\mu_*^{\otimes N})}{N}}.$$

*Proof.* We here decompose the error as follows.

$$\mathbb{E}_{\mathbf{X}\sim\mu^{(N)}}\left[\left(\mathbb{E}_{X\sim\rho_\mathbf{X}}[h(X,z)] - \mathbb{E}_{X\sim\mu_*}[h(X,z)]\right)^2\right]$$

$$= \frac{1}{N^2}\mathbb{E}_{\mathbf{X}\sim\mu^{(N)}}\left[\sum_{i=1}^{N}\left(h(X_i,z) - \mathbb{E}_{X\sim\mu_*}[h(X,z)]\right)^2\right]$$

$$+ \frac{1}{N^2}\mathbb{E}_{\mathbf{X}\sim\mu^{(N)}}\left[\sum_{i\neq j}\left(h(X_i,z) - \mathbb{E}_{X\sim\mu_*}[h(X,z)]\right)\right.$$

$$\left. \cdot\left(h(X_j,z) - \mathbb{E}_{X\sim\mu_*}[h(X,z)]\right)\right].$$

Using the boundedness of $h$, the first term can be upper bounded by $4R^2/N$. The second term can be evaluated as follows. Set $H(X_i) = h(X_i, z) - \mathbb{E}_{X\sim\mu_*}[h(X,z)]$. Then,

$$\mathbb{E}_{\mathbf{X}\sim\mu^{(N)}}[H(X_i)H(X_j)]$$

$$= \mathbb{E}_{(X_i,X_j)\sim\mu_{i,j}^{(N)}}[H(X_i)H(X_j)]$$

$$= \mathbb{E}_{(X_i,X_j)\sim\mu_*^{\otimes 2}}[H(X_i)H(X_j)]$$

$$+ \left(\mathbb{E}_{(X_i,X_j)\sim\mu_{i,j}^{(N)}} - \mathbb{E}_{(X_i,X_j)\sim\mu_*^{\otimes 2}}\right)[H(X_i)H(X_j)]$$

$$\leq 8R^2\mathrm{TV}(\mu_{1,2}^{(N)}, \mu_*^{\otimes 2})$$

$$\leq 4R^2\sqrt{2\mathrm{KL}(\mu_{1,2}^{(N)}\|\mu_*^{\otimes 2})},$$

where TV is the TV-norm and we used Pinsker's inequality. Applying Lemma 4.1 with $s = 2$, we finish the proof. □

In the proof, we see that KL-divergence controls the cross term by absorbing the difference between marginal distributions $\mu_{i,j}^{(N)}$ and $\mu^{\otimes 2}$. By combining this result with the PoC for MFLD (Theorem 3.8), we arrive at the model approximation error achieved by MFLD.

**Theorem 4.3.** *Under the same conditions as in Theorem 3.8, we run MFLD in the discrete-time, with $\eta\lambda' < 1/2$ and $\mathbf{X}_0 \sim \mu_0^{\otimes N}$. Then we get*

$$\mathbb{E}_{\mathbf{X}\sim\mu_k^{(N)}}\left[\left(\mathbb{E}_{X\sim\rho_{\mathbf{X}}}[h(X,z)] - \mathbb{E}_{X\sim\mu_*}[h(X,z)]\right)^2\right]$$

$$\leq \frac{4R^2}{N} + 8R^2\sqrt{\frac{B}{\lambda N} + \frac{\delta_\eta}{\alpha\lambda^2} + \exp\left(-\alpha\lambda\eta k\right)\frac{\Delta_0^{(N)}}{\lambda}}.$$

Note that exchangeability of $\mu_k^{(N)}$ is satisfied at all iterations because of the symmetric structure of problem and initialization with respect to particles.

**Model Ensemble** We introduce the PoC-based model ensemble to further reduce the approximation error. We first train $M$ MFNNs of $N$-neurons in parallel with the same settings and obtain sets of optimized particles $\mathbf{X}_j$ ($j = 1, 2, \ldots, M$) where each $\mathbf{X}_j = (X_j^1 \ldots, X_j^N)$ represents each network and they are independent to each other. We then integrate them into a single network of $MN$-neurons as follows:

$$\frac{1}{M}\sum_{j=1}^{M}\mathbb{E}_{X\sim\rho_{\mathbf{X}_j}}[h(X,z)] = \frac{1}{MN}\sum_{j=1}^{M}\sum_{i=1}^{N}h(X_j^i, z). \tag{10}$$

Because of the independence of networks $\{\mathbf{X}_j\}_{j=1}^M$, variance reduction occurs, resulting in the improved approximation error. Indeed, by extending Proposition 4.2 into an ensemble setting (see Proposition A.1) and using PoC (Theorem 3.8), we get the following bound. The proof is deferred to Appendix A.2.

**Theorem 4.4.** *Under the same conditions as in Theorem 3.8, we run $M$-parallel MFLD in the discrete time independently, with $\eta\lambda' < 1/2$ and $\mathbf{X}_{j,0} \sim \mu_0^{\otimes N}$ ($j = 1, 2, \ldots, M$). Then*

$$\mathbb{E}_{\{\mathbf{X}_{j,k}\}_{j=1}^M}\left[\left(\frac{1}{M}\sum_{j=1}^{M}\mathbb{E}_{\rho_{\mathbf{X}_{j,k}}}[h(X,z)] - \mathbb{E}_{\mu_*}[h(X,z)]\right)^2\right]$$

$$\leq \frac{4R^2}{MN} + \frac{8R^2}{M}\sqrt{\frac{B}{\lambda N} + \frac{\delta_\eta}{\alpha\lambda^2} + \exp(-\alpha\lambda\eta k)\frac{\Delta_0^{(N)}}{\lambda}}$$

$$+ 2R^2\left(\frac{B}{\lambda N} + \frac{\delta_\eta}{\alpha\lambda^2} + \exp(-\alpha\lambda k)\frac{\Delta_0^{(N)}}{\lambda}\right),$$

*where $\mathbf{X}_{j,k}$ is the particles at $k$-iteration for $j$-th network.*

For simplicity, we consider the bound $\frac{4R^2}{MN} + \frac{8R^2}{M}\sqrt{\frac{B}{\lambda N}} + \frac{2R^2 B}{\lambda N}$ obtained in the limit $k \to \infty$, $\eta \to 0$. This bound indicates that increasing $M$ offers better scalability than increasing $N$, as long as the second term dominates. In fact, under the constraint $MN = \Theta(K)$, where $K$ denotes the

total number of neurons, the bound suggests a non-trivial choice for the number of networks $M$. Rewriting the bound using $M$ and $K$, we obtain $O\left(\frac{1}{K} + \frac{1}{\sqrt{\lambda MK}} + \frac{M}{\lambda K}\right)$. This shows that $M$ induces a trade-off, and the bound achieves its minimum value of $O\left(\frac{1}{(\lambda K)^{2/3}}\right)$ when $M = \lambda^{1/3}K^{1/3}$. This phenomenon arises because training multiple networks enhances the independence among particles.

**Remark.** Our result can extend to randomly pruned networks. That is, we consider randomly pruning $(N - s)$-neurons after training the MFNN of $N$-neurons. Then, we get the counterpart of Proposition 4.2 as follows.

$$\mathbb{E}_{\mathbf{X}\sim\mu_{1:s}^{(N)}}\left[\left(\mathbb{E}_{X\sim\rho_{\mathbf{X}}}[h(X,z)] - \mathbb{E}_{X\sim\mu_*}[h(X,z)]\right)^2\right]$$

$$\leq \frac{4R^2}{s} + 8R^2\sqrt{\frac{\mathrm{KL}(\mu^{(N)}\|\mu_*^{\otimes N})}{N}},$$

where $\mu_{1:s}^{(N)}$ is the distribution of remaining neurons. Moreover, Theorem 4.4 can also extend to the ensemble model of randomly pruned networks in the same way.

### 4.2 Uniform model approximation error

We here consider uniform model approximation error over $z \in \mathcal{Z}$, which is more useful than point-wise evaluation in the machine learning scenario such as generalization analysis. The uniform bound essentially requires complexity evaluation of the model, and hence we make the additional assumption to control the complexity.

**Assumption 4.5.**

- The data domain is bounded: $\mathcal{Z} \subset [-1, 1]^d \subset \mathbb{R}^d$
- There exists $\beta > 0$ such that for any $x \in \mathbb{R}^d$, $z, z' \in \mathcal{Z}$,

$$|h(x, z) - h(x, z')| \leq \beta\|x\|_2\|z - z'\|_2.$$

For example, $h(x, z) = \frac{R}{3}(\tanh(x^{1\top}z + x^2) + 2\tanh(x^3))$, $(x^1 \in \mathbb{R}^d, (x^2, x^3) \in \mathbb{R}^2)$, used in Suzuki et al. (2023b) satisfies the above assumption with $\beta = \frac{R}{3}$ due to 1-Lipschitz continuity of $\tanh$.

Given random variables $\{\mathbf{X}_j\}_{j=1}^M$, $(\mathbf{X}_j = (X_j^1, \ldots, X_j^N))$, we consider the empirical Rademacher complexity of the function class $\mathcal{F} = \{x \mapsto h(x, z) \mid z \in \mathcal{Z}\}$:

$$\hat{\mathcal{R}}_{N,M}(\mathcal{F}) = \mathbb{E}_\sigma\left[\sup_{f\in\mathcal{F}}\left|\frac{1}{MN}\sum_{j=1}^{M}\sum_{i=1}^{N}\sigma_j^i f(X_j^i)\right|\right],$$

where the expectation is taken over the Rademacher random variables $\sigma = (\sigma_j^i)$ which are i.i.d. with the probability $\mathbb{P}[\sigma_j^i = 1] = \mathbb{P}[\sigma_j^i = -1] = 1$. Here, we utilize the uniform laws of large numbers to evaluate the approximation error of an ensemble model defined by $\mu_*^{\otimes N}$; note that the result for a single model is obtained as a special case $M = 1$.

**Lemma 4.6.** *Let $\mathbf{X}_j \sim \mu_*^{\otimes N} (j = 1, 2, \ldots, M)$ be independent random variables. For $\delta \in (0, 1)$, it follows that with high probability $1 - \delta$,*

$$\left\| \frac{1}{M} \sum_{j=1}^{M} \mathbb{E}_{\rho_{\mathbf{x}_j}}[h(X, \cdot)] - \mathbb{E}_{\mu_*}[h(X, \cdot)] \right\|_{\infty}$$
$$\leq 2\mathbb{E}_{\{\mathbf{X}_j\}_{j=1}^{M}} \left[ \hat{\mathcal{R}}_{N,M}(\mathcal{F}) \right] + R\sqrt{\frac{2 \log(1/\delta)}{MN}}.$$

*Proof.* The lemma follows directly by applying the uniform law of large numbers to the function class $\mathcal{F}$ (see, for instance, Mohri et al. (2012). □

The complexity $\mathbb{E}_{\{\mathbf{X}_j\}_{j=1}^{M}} \left[ \hat{\mathcal{R}}_{N,M}(\mathcal{F}) \right]$ can be then evaluated by Dudley's entropy integral (Lemma A.2) under Assumption 4.5 and the boundedness $|h(x, z)| \leq R$. By using the variational formulation of KL-divergence (e.g., Corollary 4.15 in Boucheron et al. (2013)), we translate the result of Lemma 4.6 into the approximation error of an ensemble model obtained by $M$ independent parallel iterates $\mathbf{X}_{j,k} \sim \mu_k^{(N)}$ $(j = 1, 2, \ldots, M)$ of MFLDs. Combining these techniques with Theorem 3.8, we conclude the following theorem.

**Theorem 4.7.** *Suppose Assumption 4.5 and the same conditions as in Theorem 3.8 hold. Run $M$-parallel MFLD in the discrete time independently, with $\eta\lambda' < 1/2$ and $\mathbf{X}_{j,0} \sim \mu_0^{\otimes N} (j = 1, 2, \ldots, M)$. Then we get*

$$\mathbb{E}_{\{\mathbf{X}_{j,k}\}} \left[ \left\| \frac{1}{M} \sum_{j=1}^{M} \mathbb{E}_{X \sim \rho_{\mathbf{x}_{j,k}}}[h(X, \cdot)] - \mathbb{E}_{X \sim \mu_*}[h(X, \cdot)] \right\|_{\infty} \right]$$
$$= \tilde{O} \left( R\sqrt{\frac{d}{MN} + \frac{dB}{\lambda N} + \frac{d\lambda}{MN(\lambda + MB)}} \right)$$
$$+ O \left( R\sqrt{\frac{d\lambda MN}{\lambda + MB}} \left( \frac{\delta_\eta}{\alpha\lambda^2} + \frac{1}{\lambda} \exp(-\alpha\lambda\eta k)\Delta_0^{(N)} \right) \right).$$

Here, the $\tilde{O}$-notation hides logarithmic factors. As for the concrete bound and proofs, see Appendix A.3. The term $\sqrt{\frac{d}{MN} + \frac{B}{\lambda N} + \frac{d\lambda}{MN(\lambda+MB)}}$ represents the particle approximation error due to finite $N$ particles, and even when $M = 1$, they improve upon the bound in Suzuki et al. (2023a;b) by removing the LSI constant $\alpha$ from the corresponding term. And the upper bound shows the improvement as $M$ increases.

## 5 Experiments

We verify the validity of our theoretical results, by conducting numerical experiments on synthetic data in both classification and regression settings using mean-field neural

networks. Finally, we further substantiate the applicability of our method on LoRA training of language models.

### 5.1 Mean-field neural networks

To demonstrate Theorem 4.7, we compute the log sup norm between the ouptuts of an (approximately) infinite width network and a merged network, both trained using noisy gradient descent. A finite-width approximation of the mean-field neural network is employed with $N = N_\infty$ while the merged network is obtained by merging $M$ different networks of $N$ neurons each. This is repeated across various values of $N$ and $M$. See Appendix B.1 for more details about our methodology.

**Classification setting** We consider the binary classification of $n$ data points generated along the perimeters of two concentric circles with radius $r_{inner}$ and $r_{outer}$, where labels are assigned according to the circle a data point belongs to.

**Regression setting** We consider regression on the $k$ multi-index problem. Each input sample $z_i = \left( z_i^1, \ldots, z_i^d \right) \in \mathbb{R}^d$ is generated uniformly within a $d$-dimensional hypersphere of radius $r$. Let $g : \mathbb{R}^d \to \mathbb{R}$ be a link function, then $y_i = g(z_i) = \frac{1}{k} \sum_{j=1}^{k} \tanh(z_i^j) \in \mathbb{R}$, where $k \leq d, k \in \mathbb{R}$.

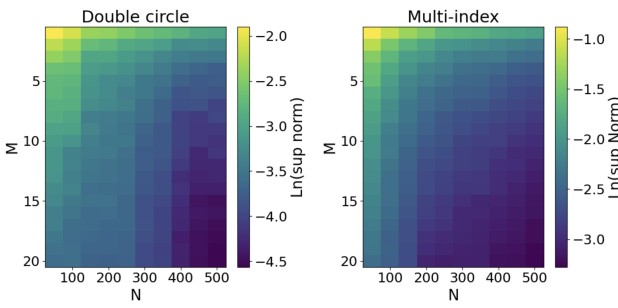

*Figure 1.* Heat maps of sup norm (in log-scale) between $N_\infty$ and merged networks when varying $M$ and $N$.

From Figure 1, we see that the merged networks converge towards the mean-field limit as $M$ and $N$ increase. This aligns with our theoretical findings which suggests that the sup norm of the approximation error decreases when more particles are added (ensembling in our experimental set-up) or increasing the ensemble size. See Appendix B.2 for supplementary experiments.

### 5.2 LoRA for finetuning language models

Beyond the scope of the theory, we empirically verify the applicability of our ensemble technique to finetuning language models using LoRA. Given a pre-trained parameter $W_0 \in \mathbb{R}^{k \times d}$ of a linear layer, LoRA introduces low-rank

*Table 1.* Accuracy comparison of LoRA and PoC-based merging for finetuning Llama models.

| Model | Method | SIQA | PIQA | WinoGrande | OBQA | ARC-c | ARC-e | BoolQ | HellaSwag | Ave. |
|---|---|---|---|---|---|---|---|---|---|---|
| Llama2 7B | LoRA (32, best) | 79.48 | 82.43 | 81.77 | 80.60 | 67.75 | 80.47 | 70.37 | 86.67 | 78.69 |
| | LoRA (256) | 69.95 | 69.69 | 69.61 | 61.40 | 47.44 | 61.15 | 63.73 | 47.27 | 61.28 |
| | **PoC merge** | 81.17 | 84.60 | 85.16 | 86.60 | 72.53 | 86.62 | 72.45 | 92.79 | 82.74 |
| Llama3 8B | LoRA (32, best) | 81.22 | 89.50 | 86.74 | 86.00 | 79.86 | 90.53 | 72.91 | 95.34 | 85.26 |
| | LoRA (256) | 81.06 | 87.60 | 87.61 | 84.60 | 78.92 | 90.06 | 75.11 | 94.98 | 84.99 |
| | **PoC merge** | 82.04 | 89.39 | 89.27 | 89.20 | 83.28 | 92.30 | 76.33 | 96.58 | 87.30 |

matrices $\mathbf{A} \in \mathbb{R}^{N \times d}$ and $\mathbf{B} \in \mathbb{R}^{k \times N}$, and represents the fine-tuned parameter as $W_0 + \gamma \Delta W = W_0 + \gamma \mathbf{B} \mathbf{A}$ ($\gamma > 0$). Then, only $\mathbf{A}$ and $\mathbf{B}$ are optimized, leaving $W_0$ frozen. Using the expression $\mathbf{A}^\top = (a^1, \ldots, a^N)$ ($a^i \in \mathbb{R}^d$) and $\mathbf{B} = (b^1, \ldots, b^N)$ ($b^i \in \mathbb{R}^k$), we can reformalize LoRA parameter $\gamma \mathbf{B} \mathbf{A}$ with $\gamma = 1/N$ as the MFNN: $\mathbb{R}^d \ni z \to \frac{1}{N} \sum_{i=1}^N h((a^i, b^i), z) \in \mathbb{R}^k$ where $h((a^i, b^i), z) = b^i a^{i\top} z$. Therefore, we can apply PoC-based model ensemble for LoRA parameters. Note that the ensemble model is reduced to a single ($k \times d$)-matrix $\Delta W$ due to the linearity of the activation function, and therefore it does not require additional memory and time for inference.

We use commonsense reasoning datasets (Hu et al., 2023): SIQA, PIQA, WinoGrande, OBQA, ARC-c, ARC-e, BoolQ, and HellaSwag, and use language models: Llama2-7B (Touvron et al., 2023) and Llama3-8B (Dubey et al., 2024). We first optimize the multiple LoRA parameters $\{(\mathbf{A}_j, \mathbf{B}_j)\}_{j=1}^M$ using noisy AdamW where $\sqrt{2\lambda \eta_k} \xi_k$ (step-size $\eta_k$, standard Gaussian noise $\xi_k$) is added to each parameter update of AdamW (Loshchilov & Hutter, 2019). Then, we merge them into $\Delta W = \frac{1}{MN} \sum_{i,j} b_j^i a_j^{i\top} \in \mathbb{R}^{k \times d}$. Hyperparameters are set to $N = 32, M = 8, \lambda = 10^{-5}$ and the number of epochs is 3. In Table 1, we compare the accuracy of the merged parameter with the base parmeters of LoRA. For LoRA, the row "LoRA (32, best)" presents the best result achieved among eight different LoRA parameters based on the average accuracy across all datasets. For both models, we observed that PoC-based model merging significantly improves the finetuning performance. For comparison under the same computational budget $MN = 256$, we report the results obtained by LoRA with $M = 1$ and $N = 256$ in the row "LoRA (256)". We also verify the performance using one-epoch training to examine the effect of Gaussian noise in parameter updates. See Appendix B.3 for more details.

## Conclusion and Discussion

We established an improved PoC for MFLD that accelerates optimization speed in Nitanda (2024); Chewi et al. (2024) while achieving the same particle complexity $O(1/N)$. We then translated this result into model approximation error

bounds, and derived a PoC-based model ensemble method with an empirical verification. Moreover, we substantiated its applicability to fine-tuning language models using LoRA.

One limitation of our theory is that it cannot explain the asymptotic behavior as $\lambda \to 0$. This is also the case in previous work since the optimization speed essentially slows down exponentially, which is inevitable in general as discussed in the literature. However, there might be room to tighten the particle approximation term $\frac{B}{\lambda N}$ with respect to $\lambda$ in the model approximation bounds. This term arises from the KL-divergence, which essentially controls the correlation among particles, as seen in the proof of Proposition 4.2. However, KL-divergence may be excessive for this purpose. Therefore, one interesting future direction is to utilize a PoC with respect to a smaller metric that alleviates the dependence on $\lambda$.

## Acknowledgements

This research is supported by the National Research Foundation, Singapore, Infocomm Media Development Authority under its Trust Tech Funding Initiative, and the Ministry of Digital Development and Information under the AI Visiting Professorship Programme (award number AIVP-2024-004). Any opinions, findings and conclusions or recommendations expressed in this material are those of the author(s) and do not reflect the views of National Research Foundation, Singapore, Infocomm Media Development Authority, and the Ministry of Digital Development and Information. TS was partially supported by JSPS KAKENHI (24K02905) and JST CREST (JPMJCR2115).

## Impact Statement

This paper presents work whose goal is to advance the field of Machine Learning. There are many potential societal consequences of our work, none which we feel must be specifically highlighted here.

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

# A Proofs

## A.1 Propagation of chaos for MFLD (Section 3)

*Proof of Lemma 3.6.* The first equality of the assertion was proved by Nitanda (2024). We here prove the inequality by utilizing the argument of the conditional and marginal distribution of $\mu^{(N)}$ (Chen et al., 2022).

By the convexity, we have that for any $\mu \in \mathcal{P}_2(\mathbb{R}^d)$,

$$\mathbb{E}_{\mathbf{X}\sim\mu^{(N)}}\left[F(\rho_{\mathbf{X}}) + \left\langle \frac{\delta F(\rho_{\mathbf{X}})}{\delta\mu}, \mu - \rho_{\mathbf{X}} \right\rangle\right] + \lambda\mathrm{Ent}(\mu) \leq \mathcal{L}(\mu). \tag{11}$$

To further evaluate the lower-bound, we begin with the following equation.

$$-\lambda \log Z(\mathbf{x}^{-i}) = \min_{\mu\in\mathcal{P}_2(\mathbb{R}^d)} \left\{ N\mathbb{E}_{X\sim\mu}[F(\rho_{X\cup\mathbf{x}^{-i}})] + \lambda\mathrm{Ent}(\mu) \right\}, \tag{12}$$

where $Z(\mathbf{x}^{-i}) = \int \exp\left(-\frac{N}{\lambda}F(\rho_{x\cup\mathbf{x}^{-i}})\right) \mathrm{d}x$ is the normalizing constant of $\nu_{i|-i}(\cdot|\mathbf{x}^{-i})$.

This equality is confirmed by reformulating the objective of (12) as follows:

$$N\mathbb{E}_{X\sim\mu}[F(\rho_{x\cup\mathbf{X}^{-i}})] + \lambda\mathrm{Ent}(\mu) = \lambda\mathrm{KL}(\mu\|\nu_{i|-i}(\cdot|\mathbf{X}^{-i})) - \lambda\log Z(\mathbf{X}^{-i}).$$

Then, the lower-bound in Eq. (11) can be evaluated as follows:

$$\mathbb{E}_{\mathbf{X}\sim\mu^{(N)}}\left[F(\rho_{\mathbf{X}}) + \left\langle \frac{\delta F(\rho_{\mathbf{X}})}{\delta\mu}, \mu - \rho_{\mathbf{X}} \right\rangle\right] + \lambda\mathrm{Ent}(\mu)$$

$$= \mathbb{E}_{\mathbf{X}\sim\mu^{(N)}}\left[F(\rho_{\mathbf{X}}) + \frac{1}{N}\sum_{i=1}^{N}\mathbb{E}_{X\sim\mu}\left[\frac{\delta F(\rho_{\mathbf{X}})}{\delta\mu}(X) - \frac{\delta F(\rho_{\mathbf{X}})}{\delta\mu}(X^i)\right]\right] + \lambda\mathrm{Ent}(\mu)$$

$$= \sum_{i=1}^{N}\mathbb{E}_{\mathbf{X}\sim\mu^{(N)}}\mathbb{E}_{X\sim\mu}\left[F(\rho_{\mathbf{X}}) + \left\langle \frac{\delta F(\rho_{\mathbf{X}})}{\delta\mu}, \rho_{X\cup\mathbf{X}^{-i}} - \rho_{\mathbf{X}} \right\rangle\right] - (N-1)\mathbb{E}_{\mathbf{X}\sim\mu^{(N)}}[\mathcal{F}(\rho_{\mathbf{X}})] + \lambda\mathrm{Ent}(\mu)$$

$$= \sum_{i=1}^{N}\mathbb{E}_{\mathbf{X}\sim\mu^{(N)}}\mathbb{E}_{X\sim\mu}\left[F(\rho_{X\cup\mathbf{X}^{-i}}) - B_F(\rho_{X\cup\mathbf{X}^{-i}}, \rho_{\mathbf{X}})\right] - (N-1)\mathbb{E}_{\mathbf{X}\sim\mu^{(N)}}[\mathcal{F}(\rho_{\mathbf{X}})] + \lambda\mathrm{Ent}(\mu)$$

$$\geq -\frac{\lambda}{N}\sum_{i=1}^{N}\mathbb{E}_{\mathbf{X}\sim\mu^{(N)}}[\log Z(\mathbf{X}^{-i})] - \frac{B}{N} - (N-1)\mathbb{E}_{\mathbf{X}\sim\mu^{(N)}}[\mathcal{F}(\rho_{\mathbf{X}})], \tag{13}$$

where we used Assumption 3.4 for the last inequality.

We consider the following decomposition:

$$\frac{\lambda}{N}\sum_{i=1}^{N}\mathbb{E}_{\mathbf{X}\sim\mu^{(N)}}\left[\mathrm{KL}(\mu_{i|-i}^{(N)}(\cdot|\mathbf{X}^{-i})\|\nu_{i|-i}(\cdot|\mathbf{X}^{-i}))\right]$$

$$= \frac{\lambda}{N}\sum_{i=1}^{N}\mathbb{E}_{\mathbf{X}\sim\mu^{(N)}}\left[\mathrm{Ent}(\mu_{i|-i}^{(N)}(\cdot|\mathbf{X}^{-i}))\right]$$

$$+ \sum_{i=1}^{N}\mathbb{E}_{\mathbf{X}\sim\mu^{(N)}}\left[\int F(\rho_{x\cup\mathbf{X}^{-i}})\mu_{i|-i}^{(N)}(\mathrm{d}x|\mathbf{X}^{-i})\right]$$

$$+ \frac{\lambda}{N}\sum_{i=1}^{N}\mathbb{E}_{\mathbf{X}\sim\mu^{(N)}}\left[\log Z(\mathbf{X}^{-i})\right]. \tag{14}$$

The first term can be bounded by the well-known property of entropy.

$$\frac{\lambda}{N}\sum_{i=1}^{N}\mathbb{E}_{\mathbf{X}\sim\mu^{(N)}}\left[\mathrm{Ent}(\mu_{i|-i}^{(N)}(\cdot|\mathbf{X}^{-i}))\right] \geq \frac{\lambda}{N}\mathrm{Ent}(\mu^{(N)}). \tag{15}$$

The second term can be simply written as follows.

$$\sum_{i=1}^{N} \mathbb{E}_{\mathbf{X} \sim \mu^{(N)}} \left[ \int F(\rho_{x \cup \mathbf{X}^{-i}}) \mu_{i|-i}^{(N)}(\mathrm{d}x|\mathbf{X}^{-i}) \right] = N \mathbb{E}_{\mathbf{X} \sim \mu^{(N)}}[F(\rho_{\mathbf{X}})]. \tag{16}$$

Combining (13) (14), (16), and (15), we get

$$\mathbb{E}_{\mathbf{X} \sim \mu^{(N)}} \left[ F(\rho_{\mathbf{X}}) + \left\langle \frac{\delta F(\rho_{\mathbf{X}})}{\delta \mu}, \mu - \rho_{\mathbf{X}} \right\rangle \right] + \lambda \mathrm{Ent}(\mu)$$

$$\geq -\frac{\lambda}{N} \sum_{i=1}^{N} \mathbb{E}_{\mathbf{X} \sim \mu^{(N)}} \left[ \mathrm{KL}(\mu_{i|-i}^{(N)}(\cdot|\mathbf{X}^{-i}) \| \nu_{i|-i}(\cdot|\mathbf{X}^{-i})) \right] + \frac{\lambda}{N} \mathrm{Ent}(\mu^{(N)}) - \frac{B}{N} + \mathbb{E}_{\mathbf{X} \sim \mu^{(N)}}[\mathcal{F}(\rho_{\mathbf{X}})]$$

$$\geq -\frac{B}{N} + \frac{1}{N} \mathcal{L}^{(N)}(\mu^{(N)}) - \frac{\lambda}{N} \sum_{i=1}^{N} \mathbb{E}_{\mathbf{X} \sim \mu^{(N)}} \left[ \mathrm{KL}(\mu_{i|-i}^{(N)}(\cdot|\mathbf{X}^{-i}) \| \nu_{i|-i}(\cdot|\mathbf{X}^{-i})) \right]. \tag{17}$$

This concludes

$$\frac{\lambda}{N} \sum_{i=1}^{N} \mathbb{E}_{\mathbf{X} \sim \mu^{(N)}} \left[ \mathrm{KL}(\mu_{i|-i}^{(N)}(\cdot|\mathbf{X}^{-i}) \| \nu_{i|-i}(\cdot|\mathbf{X}^{-i})) \right] \geq -\frac{B}{N} + \frac{1}{N} \mathcal{L}^{(N)}(\mu^{(N)}) - \mathcal{L}(\mu).$$

We finish the proof by setting $\mu = \mu_*$. $\qquad\square$

*Proof of Lemma 3.7.* We denote by $\mu_{-i}^{(N)}$ the marginal distribution over $\mathbf{X}^{-i}$. It holds that

$$\mathbb{E}_{\mathbf{X} \sim \mu^{(N)}} \left[ \left\| \nabla \log \frac{\mathrm{d}\mu^{(N)}}{\mathrm{d}\mu_*^{(N)}}(\mathbf{X}) \right\|_2^2 \right]$$

$$= \sum_{i=1}^{N} \mathbb{E}_{\mathbf{X} \sim \mu^{(N)}} \left[ \left\| \nabla_{x^i} \log \frac{\mathrm{d}\mu^{(N)}}{\mathrm{d}\mathbf{x}}(\mathbf{X}) + \frac{N}{\lambda} \nabla_{x^i} F(\rho_{\mathbf{X}}) \right\|_2^2 \right]$$

$$= \sum_{i=1}^{N} \mathbb{E}_{\mathbf{X}^{-i} \sim \mu_{-i}^{(N)}} \left[ \mathbb{E}_{X^i \sim \mu_{i|-i}^{(N)}(\cdot|\mathbf{X}^{-i})} \left[ \left\| \nabla_{x^i} \log \frac{\mathrm{d}\mu^{(N)}}{\mathrm{d}\mathbf{x}}(\mathbf{X}) + \frac{N}{\lambda} \nabla_{x^i} F(\rho_{\mathbf{X}}) \right\|_2^2 \right] \right]. \tag{18}$$

We write $p_{-i}(\mathbf{x}^{-i}) = \frac{\mathrm{d}\mu_{-i}^{(N)}}{\mathrm{d}\mathbf{x}^{-i}}(\mathbf{x}^{-i})$ and $p_{i|-i}(x|\mathbf{x}^{-i}) = \frac{\mathrm{d}\mu_{i|-i}^{(N)}(\cdot|\mathbf{x}^{-i})}{\mathrm{d}x}(x)$. Since $\frac{\mathrm{d}\mu^{(N)}}{\mathrm{d}\mathbf{x}}(\mathbf{x}) = p_{-i}(\mathbf{x}^{-i})p_{i|-i}(x^i|\mathbf{x}^{-i})$, we get the following equation:

$$\nabla_{x^i} \log \frac{\mathrm{d}\mu^{(N)}}{\mathrm{d}\mathbf{x}}(\mathbf{x}) = \frac{\nabla_{x^i}(p_{-i}(\mathbf{x}^{-i})p_{i|-i}(x^i|\mathbf{x}^{-i}))}{p_{-i}(\mathbf{x}^{-i})p_{i|-i}(x^i|\mathbf{x}^{-i})} = \frac{\nabla_{x^i} p_{i|-i}(x^i|\mathbf{x}^{-i})}{p_{i|-i}(x^i|\mathbf{x}^{-i})} = \nabla \log p_{i|-i}(x^i|\mathbf{x}^{-i}).$$

Hence, Eq. (18) can be further bounded by the LSI on the conditional Gibbs distribution (Assumption 3.2) as follows:

$$\sum_{i=1}^{N} \mathbb{E}_{\mathbf{X}^{-i} \sim \mu_{-i}^{(N)}} \left[ \mathbb{E}_{X^i \sim \mu_{i|-i}^{(N)}(\cdot|\mathbf{X}^{-i})} \left[ \left\| \nabla \log p_{i|-i}(X^i|\mathbf{X}^{-i}) + \frac{N}{\lambda} \nabla_{x^i} F(\rho_{\mathbf{X}}) \right\|_2^2 \right] \right]$$

$$= \sum_{i=1}^{N} \mathbb{E}_{\mathbf{X}^{-i} \sim \mu_{-i}^{(N)}} \left[ \mathbb{E}_{X^i \sim \mu_{i|-i}^{(N)}(\cdot|\mathbf{X}^{-i})} \left[ \left\| \nabla \log \frac{\mathrm{d}\mu_{i|-i}^{(N)}}{\mathrm{d}\nu_{i|-i}}(X^i|\mathbf{X}^{-i}) \right\|_2^2 \right] \right]$$

$$\geq 2\alpha \sum_{i=1}^{N} \mathbb{E}_{\mathbf{X}^{-i} \sim \mu_{-i}^{(N)}} \left[ \mathrm{KL}(\mu_{i|-i}^{(N)}(\cdot|\mathbf{X}^{-i}) \| \nu_{i|-i}(\cdot|\mathbf{X}^{-i})) \right]. \tag{19}$$

Combining this inequality and Lemma 3.6, we get

$$\mathbb{E}_{\mathbf{x}\sim\mu^{(N)}}\left[\left\|\nabla\log\frac{\mathrm{d}\mu^{(N)}}{\mathrm{d}\mu_*^{(N)}}(\mathbf{X})\right\|_2^2\right]\geq\frac{2\alpha}{\lambda}\left(-B+\mathcal{L}^{(N)}(\mu^{(N)})-N\mathcal{L}(\mu_*)\right).$$

This concludes the proof. $\qquad\square$

*Proof of Theorem 3.8.* We here prove the convergence of MFLD in the discrete-time by using the one-step interpolation argument (Nitanda, 2024; Suzuki et al., 2023a).

We construct the one-step interpolation for $k$-th iteration: $X_{k+1}^i = X_k^i - \eta\nabla\frac{\delta F(\rho_{\mathbf{X}_k})}{\delta\mu}(X_k^i) + \sqrt{2\lambda\eta}\xi_k^i$, $(i\in\{1,2,\ldots,d\})$. as follows: for $i\in\{1,2,\ldots,d\}$,

$$\mathrm{d}Y_t^i = -\nabla\frac{\delta F(\rho_{\mathbf{Y}_0})}{\delta\mu}(Y_0^i)\mathrm{d}t + \sqrt{2\lambda}\mathrm{d}W_t, \tag{20}$$

where $\mathbf{Y}_0 = (Y_0^1,\ldots,Y_0^d) = (X_k^1,\ldots,X_k^d)$ and $W_t$ is the standard Brownian motion in $\mathbb{R}^d$ with $W_0 = 0$. We denote by $\nu_t$ the distributions of $\mathbf{Y}_t$. Then, $\nu_0 = \mu_k^{(N)}(=\mathrm{Law}(\mathbf{X}_k))$, $\nu_\eta = \mu_{k+1}^{(N)}(=\mathrm{Law}(\mathbf{X}_{k+1}))$ (i.e., $\mathbf{Y}_\eta \overset{\mathrm{d}}{=} \mathbf{X}_{k+1}$). In this proof, we identify the probability distribution with its density function with respect to the Lebesgure measure for notational simplicity. For instance, we denote by $\mu_*^{(N)}(\mathbf{y})$ the density of $\mu_*^{(N)}$.

By the proof of Theorem 2 in Nitanda (2024), we see for $t\in[0,\eta]$,

$$\frac{\mathrm{d}\mathcal{L}^{(N)}}{\mathrm{d}t}(\nu_t) \leq -\frac{\lambda^2}{2}\int\nu_t(\mathbf{y})\left\|\nabla\log\frac{\nu_t}{\mu_*^{(N)}}(\mathbf{y})\right\|_2^2\mathrm{d}\mathbf{y} + N\delta_\eta, \tag{21}$$

where $\delta_\eta = 8\eta(C_2^2+\lambda'^2)(\eta C_1^2+\lambda d) + 32\eta^2\lambda'^2(C_2^2+\lambda'^2)\left(\frac{1}{N}\mathbb{E}\left[\|\mathbf{X}_0\|_2^2\right] + \frac{1}{\lambda'}\left(\frac{C_1^2}{4\lambda'}+\lambda d\right)\right)$.

Combining Lemma 3.7 with the above inequality, we get

$$\frac{\mathrm{d}\mathcal{L}^{(N)}}{\mathrm{d}t}(\nu_t) \leq -\alpha\lambda\left(\mathcal{L}^{(N)}(\nu_t)-N\mathcal{L}(\mu_*)-B\right) + N\delta_\eta.$$

$$\iff \frac{\mathrm{d}}{\mathrm{d}t}\left(\mathcal{L}^{(N)}(\nu_t)-N\mathcal{L}(\mu_*)-B-\frac{N\delta_\eta}{\alpha\lambda}\right) \leq -\alpha\lambda\left(\mathcal{L}^{(N)}(\nu_t)-N\mathcal{L}(\mu_*)-B-\frac{N\delta_\eta}{\alpha\lambda}\right).$$

Noting $\nu_\eta = \mu_{k+1}^{(N)}$ and $\nu_0 = \mu_k^{(N)}$, the Grönwall's inequality leads to

$$\mathcal{L}^{(N)}(\mu_{k+1}^{(N)})-N\mathcal{L}(\mu_*)-B-\frac{N\delta_\eta^{(N)}}{\alpha\lambda} \leq \exp(-\alpha\lambda\eta)\left(\mathcal{L}^{(N)}(\mu_k^{(N)})-N\mathcal{L}(\mu_*)-B-\frac{N\delta_\eta^{(N)}}{\alpha\lambda}\right).$$

This inequality holds at every iteration of (20). Hence, we arrive at the desired result,

$$\frac{1}{N}\mathcal{L}^{(N)}(\mu_k^{(N)})-\mathcal{L}(\mu_*) \leq \frac{B}{N}+\frac{\delta_\eta^{(N)}}{\alpha\lambda} + \exp(-\alpha\lambda\eta k)\left(\frac{1}{N}\mathcal{L}^{(N)}(\mu_0^{(N)})-\mathcal{L}(\mu_*)-\frac{B}{N}-\frac{\delta_\eta^{(N)}}{\alpha\lambda}\right)$$

$$\leq \frac{B}{N}+\frac{\delta_\eta^{(N)}}{\alpha\lambda} + \exp(-\alpha\lambda\eta k)\left(\frac{1}{N}\mathcal{L}^{(N)}(\mu_0^{(N)})-\mathcal{L}(\mu_*)\right).$$

$\qquad\square$

## A.2 Point-wise model approximation error (Section 4.1)

*Proof of Lemma 4.1.* It follows that by Han's inequality (Dembo et al., 1991),

$$\frac{1}{s\binom{N}{s}}\sum_{|S|=s}\int\mu_S^{(N)}(\mathrm{d}\mathbf{x}_S)\log\frac{\mathrm{d}\mu_S^{(N)}}{\mathrm{d}\mathbf{x}_S}(\mathbf{x}_S) \leq \frac{1}{N}\int\mu^{(N)}(\mathrm{d}x)\log\frac{\mathrm{d}\mu^{(N)}}{\mathrm{d}\mathbf{x}}(\mathbf{x}).$$

Moreover, we see

$$
\sum_{|S|=s} \int \mu_S^{(N)}(\mathrm{d}\mathbf{x}_S) \log \frac{\mathrm{d}\mu_*^{\otimes k}}{\mathrm{d}\mathbf{x}_S}(\mathbf{x}_S) = \sum_{|S|=s} \sum_{i \in S} \int \mu_i^{(N)}(\mathrm{d}x^i) \log \frac{\mathrm{d}\mu_*}{\mathrm{d}x}(x^i)
$$

$$
= \binom{N-1}{s-1} \sum_{i=1}^{N} \int \mu_i^{(N)}(\mathrm{d}x^i) \log \frac{\mathrm{d}\mu_*}{\mathrm{d}x}(x^i)
$$

$$
= \binom{N-1}{s-1} \int \mu^{(N)}(\mathrm{d}\mathbf{x}) \log \frac{\mathrm{d}\mu_*^{\otimes N}}{\mathrm{d}\mathbf{x}}(\mathbf{x}).
$$

Noticing $\binom{N-1}{s-1} = \frac{s}{N}\binom{N}{s}$, we conclude the first statement which immediately implies the second statement. $\square$

**Proposition A.1.** *Suppose $\mu^{(N)}$ is exchangeable and $\mathbf{X}_j \sim \mu^{\otimes N}$ $(j = 1, 2, \ldots, M)$. Then, it follows that for any $z \in \mathcal{Z}$,*

$$
\mathbb{E}_{\{\mathbf{X}_j\}_{j=1}^{M}} \left[ \left( \frac{1}{M} \sum_{j=1}^{M} \mathbb{E}_{\rho_{\mathbf{X}_j}}[h(X, z)] - \mathbb{E}_{\mu_*}[h(X, z)] \right)^2 \right] \leq \frac{4R^2}{NM} + \frac{8R^2}{M}\sqrt{\frac{\mathrm{KL}(\mu^{(N)}\|\mu_*^{\otimes N})}{N}} + \frac{2R^2\mathrm{KL}(\mu^{(N)}\|\mu_*^{\otimes N})}{N}.
$$

*Proof of Proposition A.1.*

$$
\mathbb{E}_{\{\mathbf{X}_j\}_{j=1}^{M}} \left[ \left( \frac{1}{M} \sum_{j=1}^{M} \mathbb{E}_{X \sim \rho_{\mathbf{X}_j}}[h(X, z)] - \mathbb{E}_{X \sim \mu_*}[h(X, z)] \right)^2 \right]
$$

$$
= \frac{1}{M^2} \mathbb{E}_{\{\mathbf{X}_j\}_{j=1}^{M}} \left[ \sum_{j=1}^{M} \left( \mathbb{E}_{X \sim \rho_{\mathbf{X}_j}}[h(X, z)] - \mathbb{E}_{X \sim \mu_*}[h(X, z)] \right)^2 \right]
$$

$$
+ \frac{1}{M^2} \mathbb{E}_{\{\mathbf{X}_j\}_{j=1}^{M}} \left[ \sum_{j \neq k} \left( \mathbb{E}_{X \sim \rho_{\mathbf{X}_j}}[h(X, z)] - \mathbb{E}_{X \sim \mu_*}[h(X, z)] \right) \left( \mathbb{E}_{X \sim \rho_{\mathbf{X}_k}}[h(X, z)] - \mathbb{E}_{X \sim \mu_*}[h(X, z)] \right) \right].
$$

Using Proposition 4.2, we can upper bound the first term by $\frac{4R^2}{Ms'} + \frac{8R^2}{M}\sqrt{\frac{\mathrm{KL}(\mu^{(N)}\|\mu_*^{\otimes N})}{N}}$. The second term can be evaluated as follows. Set $H(\mathbf{X}_j) = \mathbb{E}_{X \sim \mu_{\mathbf{X}_j}}[h(X, z)] - \mathbb{E}_{X \sim \mu_*}[h(X, z)]$. Then for $j \neq k$,

$$
\mathbb{E}_{\{\mathbf{X}_j\}_{j=1}^{M}}[H(\mathbf{X}_j)H(\mathbf{X}_k)] = \left( \mathbb{E}_{\mathbf{X}_j}[H(\mathbf{X}_j)] \right)^2
$$

$$
= \left( \mathbb{E}_{\mathbf{X}_j} \left[ \frac{1}{s'} \sum_{i=1}^{s'} h(X_j^i, z) \right] - \mathbb{E}_{X \sim \mu_*}[h(X, z)] \right)^2
$$

$$
= \left( \mathbb{E}_{X \sim \mu_1^{(N)}}[h(X, z)] - \mathbb{E}_{X \sim \mu_*}[h(X, z)] \right)^2
$$

$$
\leq 4R^2 \mathrm{TV}^2(\mu_1^{(N)}, \mu_*)
$$

$$
\leq 2R^2 \mathrm{KL}(\mu_1^{(N)}\|\mu_*)
$$

$$
\leq \frac{2R^2}{N} \mathrm{KL}(\mu^{(N)}\|\mu_*^{\otimes N}).
$$

This concludes the proof. $\square$

### A.3 Uniform model approximation error (Section 4.2)

We evaluate the empirical Rademacher complexity $\hat{\mathcal{R}}_{N,M}(\mathcal{F})$ by using Dudley's entropy integral. We define the metric $\|f\|_{N,M,2} = \sqrt{\frac{1}{MN} \sum_{j=1}^{M} \sum_{i=1}^{N} |f(X_j^i)|^2}$. We denote by $\mathcal{N}(\mathcal{F}, \epsilon, \|\cdot\|_{N,M,2})$ the $\epsilon$-covering number of $\mathcal{F}$ with respect to the $\|\cdot\|_{N,M,2}$-norm.

**Lemma A.2** (Dudley's entropy integral). *Given a function class $\mathcal{F}$ on $\mathbb{R}^d$, we suppose $R = \sup_{f \in \mathcal{F}} \|f\|_{N,M,2} < \infty$. Then,*

$$\hat{\mathcal{R}}_{N,M}(\mathcal{F}) \leq \inf_{\delta > 0} \left\{ 4\delta + \frac{12}{\sqrt{MN}} \int_{\delta}^{R} \sqrt{\log 2\mathcal{N}(\mathcal{F}, \epsilon, \|\cdot\|_{N,M,2})} d\epsilon \right\}.$$

**Proposition A.3.** *Suppose Assumption 4.5 holds and $\mathbf{X}_j \sim \mu^{(N)}$ $(j = 1, 2, \ldots, M)$ are independent. Then, we get*

$$\mathbb{E}_{\{\mathbf{X}_j\}_{j=1}^M} \left[ \hat{\mathcal{R}}_{N,M}(\mathcal{F}) \right] \leq 4R\sqrt{\frac{d}{MN}} + 12R\sqrt{\frac{1}{MN} \left( \log 2 + d\log\left( 1 + 2\beta M R^{-1}\sqrt{MN d^{-1}} \mathbb{E}_{\mathbf{X} \sim \mu^{(N)}}[\|\mathbf{X}\|_2] \right) \right)}.$$

*Proof.* Since $\|f\|_{N,M,2} \leq \|f\|_{N,M,\infty} = \max_{i,j} |f(X_j^i)|$, it is sufficient evaluate the $\epsilon$-covering number of $\mathcal{F}$ with respect to $\|\cdot\|_{N,M,\infty}$. We write $r = \max_{i,j} \|X_j^i\|_2$. By Assumption 4.5, for any $z, z' \in \mathcal{Z}$,

$$\max_{i,j} |h(X_j^i, z) - h(X_j^i, z')| \leq \max_{i,j} \beta \|X_j^i\|_2 \|z - z'\|_2 = \beta r \|z - z'\|_2,$$

we see $\mathcal{N}(\mathcal{F}, \epsilon, \|\cdot\|_{N,M,\infty}) \leq \mathcal{N}(\mathcal{Z}, \epsilon/(\beta r), \|\cdot\|_2) = \left( 1 + \frac{2\beta r}{\epsilon} \right)^d$.

Therefore, by Lemma A.2 with $\delta = R\sqrt{d(MN)^{-1}}$, we get

$$\hat{\mathcal{R}}_{N,M}(\mathcal{F}) \leq 4R\sqrt{\frac{d}{MN}} + 12R\sqrt{\frac{1}{MN} \log 2\mathcal{N}(\mathcal{F}, R\sqrt{d(MN)^{-1}}, \|\cdot\|_{N,M,\infty})}$$

$$= 4R\sqrt{\frac{d}{MN}} + 12R\sqrt{\frac{1}{MN} \left( \log 2 + d\log\left( 1 + 2\beta r R^{-1}\sqrt{MN d^{-1}} \right) \right)}$$

$$\leq 4R\sqrt{\frac{d}{MN}} + 12R\sqrt{\frac{1}{MN} \left( \log 2 + d\log\left( 1 + 2\beta R^{-1}\sqrt{MN d^{-1}} \sum_{j=1}^{M} \|\mathbf{X}_j\|_2 \right) \right)},$$

where we used $r \leq \sum_{j=1}^{M} \|\mathbf{X}_j\|_2$. Finally, Jensen's inequality yields

$$\mathbb{E}_{\{\mathbf{X}_j\}_{j=1}^M} \left[ \hat{\mathcal{R}}_{N,M}(\mathcal{F}) \right] \leq 4R\sqrt{\frac{d}{MN}} + 12R\sqrt{\frac{1}{MN} \left( \log 2 + d\log\left( 1 + 2\beta M R^{-1}\sqrt{MN d^{-1}} \mathbb{E}_{\mathbf{X} \sim \mu^{(N)}}[\|\mathbf{X}\|_2] \right) \right)}.$$

$\square$

Here, we give the complete version of the uniform model approximation bound.

**Theorem A.4** (Complete version of Theorem 4.7). *Suppose Assumption 4.5 and the same conditions as in Theorem 3.8 hold. Run $M$-parallel MFLD in the discrete time independently, with $\eta\lambda' < 1/2$ and $\mathbf{X}_{j,0} \sim \mu_0^{\otimes N}$ $(j = 1, 2, \ldots, M)$. Then,*

$$\mathbb{E}_{\{\mathbf{X}_{j,k}\}} \left[ \left\| \frac{1}{M} \sum_{j=1}^{M} \mathbb{E}_{X \sim \rho_{\mathbf{x}_{j,k}}} [h(X, \cdot)] - \mathbb{E}_{X \sim \mu_*}[h(X, \cdot)] \right\|_{\infty} \right]$$

$$\leq \frac{5CR}{4}\sqrt{\frac{d}{MN} + \frac{dB}{\lambda N}} + CR\sqrt{\frac{d\lambda}{MN(\lambda + MB)}} \log\left( C'\sqrt{(\lambda + MB)\frac{\pi}{\lambda}} \right)$$

$$+ CR\sqrt{\frac{d\lambda MN}{\lambda + MB}} \left( \frac{\delta_\eta}{\alpha\lambda^2} + \frac{1}{\lambda}\exp(-\alpha\lambda\eta k)\Delta_0^{(N)} \right),$$

*where $C' = 1 + 2\beta M R^{-1}\sqrt{MN d^{-1}} \mathbb{E}_{\mathbf{X} \sim \mu_*^{\otimes N}}[\|\mathbf{X}\|_2]$.*

*Proof.* For $\mathbf{x}_1, \ldots, \mathbf{x}_M \in \mathbb{R}^{dN}$, we set $g(\mathbf{x}_1, \ldots, \mathbf{x}_M) = \sup_{z \in \mathcal{Z}} \left| \frac{1}{M} \sum_{j=1}^{M} \mathbb{E}_{X \sim \rho_{\mathbf{x}_j}}[h(X, z)] - \mathbb{E}_{X \sim \mu_*}[h(X, z)] \right|$. By the variational formulation of KL-divergence (e.g., Corollary 4.15 in Boucheron et al. (2013)), we get

$$
\begin{aligned}
\mathbb{E}_{\mu^{(N) \otimes M}}[g] &\leq \frac{1}{\gamma} \log \mathbb{E}_{\mu_*^{\otimes NM}}[\exp(\gamma g)] + \frac{\mathrm{KL}(\mu^{(N) \otimes M} \| \mu_*^{\otimes NM})}{\gamma} \\
&\leq \frac{1}{\gamma} \log \mathbb{E}_{\mu_*^{\otimes NM}}[\exp(\gamma g)] + \frac{M \mathrm{KL}(\mu^{(N)} \| \mu_*^{\otimes N})}{\gamma}
\end{aligned}
\tag{22}
$$

For independent random variables $\mathbf{X}_j \sim \mu_*^{\otimes N} (j = 1, 2, \ldots, M)$, by Lemma 4.6 and A.3, it follows that with high probability $1 - \delta$,

$$
\begin{aligned}
&g(\mathbf{X}_1, \ldots, \mathbf{X}_M) \\
&\leq 2\mathbb{E}_{\{\mathbf{X}_j\}_{j=1}^M} \left[ \hat{\mathcal{R}}_{N,M}(\mathcal{F}) \right] + R\sqrt{\frac{2\log(1/\delta)}{MN}} \\
&\leq 8R\sqrt{\frac{d}{MN}} + 24R\sqrt{\frac{1}{MN} \left( \log 2 + d\log\left( 1 + 2\beta MR^{-1}\sqrt{MNd^{-1}}\mathbb{E}_{\mathbf{X} \sim \mu_*^{\otimes N}}[\|\mathbf{X}\|_2] \right) \right)} + R\sqrt{\frac{2\log(1/\delta)}{MN}} \\
&\leq CR\sqrt{\frac{d\log(C'/\delta)}{MN}},
\end{aligned}
$$

where $C$ is a uniform constant and $C' = 1 + 2\beta MR^{-1}\sqrt{MNd^{-1}}\mathbb{E}_{\mathbf{X} \sim \mu_*^{\otimes N}}[\|\mathbf{X}\|_2]$. This means

$$
\begin{aligned}
&\mathbb{P}_{\mu_*^{\otimes NM}} \left[ g(\mathbf{X}_1, \ldots, \mathbf{X}_M) > CR\sqrt{\frac{d\log(C'/\delta)}{MN}} \right] \leq \delta \\
\Longleftrightarrow \quad &\mathbb{P}_{\mu_*^{\otimes NM}} [g(\mathbf{X}_1, \ldots, \mathbf{X}_M) > t] \leq C' \exp\left( -\frac{MNt^2}{dC^2R^2} \right) \\
\Longleftrightarrow \quad &\mathbb{P}_{\mu_*^{\otimes NM}} \left[ g(\mathbf{X}_1, \ldots, \mathbf{X}_M) > \frac{1}{\gamma}\log t \right] \leq C' \exp\left( -\frac{MN(\log t)^2}{dC^2R^2\gamma^2} \right).
\end{aligned}
$$

Using this tail bound,

$$
\begin{aligned}
\mathbb{E}_{\mu_*^{\otimes NM}}[\exp(\gamma g)] &= \int_0^\infty \mathbb{P}_{\mu_*^{\otimes NM}} \left[ \exp(\gamma g(\mathbf{X}_1, \ldots, \mathbf{X}_M)) > t \right] \mathrm{d}t \\
&= \int_0^\infty \mathbb{P}_{\mu_*^{\otimes NM}} \left[ g(\mathbf{X}_1, \ldots, \mathbf{X}_M) > \frac{1}{\gamma}\log t \right] \mathrm{d}t \\
&= \int_0^\infty C' \exp\left( -\frac{MN(\log t)^2}{dC^2R^2\gamma^2} \right) \mathrm{d}t \\
&= C'CR\gamma\sqrt{\frac{\pi d}{MN}} \exp\left( \frac{dC^2R^2\gamma^2}{4MN} \right).
\end{aligned}
$$

Therefore, we get

$$
\mathbb{E}_{\mu^{(N) \otimes M}}[g] \leq \frac{dC^2R^2\gamma}{4MN} + \frac{1}{\gamma}\log\left( C'CR\gamma\sqrt{\frac{\pi d}{MN}} \right) + \frac{M\mathrm{KL}(\mu^{(N)} \| \mu_*^{\otimes N})}{\gamma}.
$$

Moreover, by applying Lemma 3.7 Theorem 3.8 to Eq. (22), we get

$$
\begin{aligned}
&\mathbb{E}_{\{\mathbf{X}_{j,k}\}} \left[ \left\| \frac{1}{M} \sum_{j=1}^{M} \mathbb{E}_{X \sim \rho_{\mathbf{x}_{j,k}}}[h(X, \cdot)] - \mathbb{E}_{X \sim \mu_*}[h(X, \cdot)] \right\|_\infty \right] \\
&\leq \frac{dC^2R^2\gamma}{4MN} + \frac{1}{\gamma}\log\left( C'CR\gamma\sqrt{\frac{\pi d}{MN}} \right) + \frac{M}{\gamma}\left( \frac{B}{\lambda} + \frac{N\delta_\eta}{\alpha\lambda^2} + \frac{N}{\lambda}\exp(-\alpha\lambda\eta k)\Delta_0^{(N)} \right).
\end{aligned}
$$

Finally, by seting $\gamma = \frac{1}{CR}\sqrt{\frac{MN}{d}\left(1 + \frac{MB}{\lambda}\right)}$, we get

$$
\mathbb{E}_{\{\mathbf{X}_{j,k}\}}\left[\left\|\frac{1}{M}\sum_{j=1}^{M}\mathbb{E}_{X\sim\rho_{\mathbf{x}_{j,k}}}[h(X,\cdot)] - \mathbb{E}_{X\sim\mu_*}[h(X,\cdot)]\right\|_{\infty}\right]
$$

$$
\leq \frac{5CR}{4}\sqrt{\frac{d}{MN} + \frac{dB}{\lambda N}} + CR\sqrt{\frac{d\lambda}{MN(\lambda + MB)}}\log\left(C'\sqrt{(\lambda + MB)\frac{\pi}{\lambda}}\right)
$$

$$
+ CR\sqrt{\frac{d\lambda MN}{\lambda + MB}}\left(\frac{\delta_\eta}{\alpha\lambda^2} + \frac{1}{\lambda}\exp(-\alpha\lambda\eta k)\Delta_0^{(N)}\right).
$$

$\square$

# B Experiments

The code used in this work will be made publicly available later.

## B.1 Pseudocode and training settings for mean-field experiments

For experiments concerning MFNNs, the output of a neuron in a two-layer MFNN is modelled by: $h(x_i, z_i) = R \tanh(x_i^3) \tanh(x_i^{1\top} z_i + x_i^2)$, where $x_i = (x_i^1, x_i^2, x_i^3) \in \mathbb{R}^{d+1+1}$ is its parameter, $z_i$ is the given input and $R$ is a scaling constant. The $\tanh$ activation function is placed on the second layer as boundedness of the model is crucial for our analysis. Noisy gradient descent is then used to train neural networks for $T$ epochs each. We omit the pseudocode for training MFNNs with MFLD since it is identical to the backpropagation with noisy gradient descent algorithm.

**Algorithm 1** Generate the double circle data: $\mathcal{D} = (z_i, y_i)_{i=1}^n$, $z_i \in \mathbb{R}^2$, $y_i \in \mathbb{R}$ before splitting it into $\mathcal{D}_{\text{train}}$ and $\mathcal{D}_{\text{test}}$. We set $n = 200$, $r_{\text{inner}} = 1$, $r_{\text{outer}} = 2$ and use an 80-20 train-test split for the data.

**Algorithm 2** Generate the $k$ multi-index data: $\mathcal{D} = (z_i, y_i)_{i=1}^n$, $z_i \in \mathbb{R}^d$, $y_i \in \mathbb{R}$. A key step is normalizing and projecting $z_i$ to the inside of a $d$-dimensional hypersphere. We set $n = 500$, $d = 100$, $r = 5$, $k = 100$ and $\bar{R} = 100$.

**Algorithm 3** Describes how we obtain and test the performance of merged MFNNs against (an approximation to) the mean-field limit by computing the sup-norm between both outputs. The relevant results are stored into a dictionary for plotting the heatmaps. The training procedure is identical for both the classification and regression problem. Let $M_{\text{max}} = 20$ and $N_{\text{list}} = \{50, 100, \ldots, 500\}$ denote the maximum number of networks to merge and list of neuron settings to train in parallel respectively. We set the hyperparameters for training as follows:

- Classification: $R = 10$, $N_\infty = 10000$, $\eta = 0.1$, $\lambda' = 0.1$, $\lambda = 0.01$, $T = 200$ and loss function: logistic loss

- Regression: $R = 10$, $N_\infty = 10000$, $\eta = 0.01$, $\lambda' = 0.1$, $\lambda = 0.01$, $T = 100$ and loss function: mean squared error

---

**Algorithm 1** Generate data points along cocentric 2D circles

**Require:** $n$, $r_{\text{inner}}$, $r_{\text{outer}}$
**Ensure:** Dataset $\mathcal{D} = \{(z_i, y_i)\}_{i=1}^n$
1: Initialize $\mathcal{D} \leftarrow \emptyset$
2: **for** $i = 1$ to $n$ **do**
3:     Sample $\theta \sim \text{Uniform}(0, 2\pi)$
4:     Sample $\xi_1, \xi_2 \sim \text{Normal}(0, 0.1)$
5:     **if** $i < n/2$ **then**
6:         $r \leftarrow r_{\text{inner}}$
7:         $y_i \leftarrow -1$
8:     **else**
9:         $r \leftarrow r_{\text{outer}}$
10:        $y_i \leftarrow +1$
11:     **end if**
12:     Compute Cartesian coordinates: $z_i = (r \cos(\theta) + \xi_1, r \sin(\theta) + \xi_2)$
13:     Add $(z_i, y_i)$ to $\mathcal{D}$
14: **end for**
15: Randomly shuffle $\mathcal{D}$
16: Split $\mathcal{D}$ into $\mathcal{D}_{\text{train}}$ (80%) and $\mathcal{D}_{\text{test}}$ (20%)
17: **return** $\mathcal{D}_{\text{train}}, \mathcal{D}_{\text{test}}$

---

---

**Algorithm 2** Generate $k$ multi-index data

---

**Require:** $n, d, r, k, \bar{R}$
**Ensure:** Dataset $\mathcal{D} = \{(z_i, y_i)\}_{i=1}^{n}$
 1: Initialize $\mathcal{D} \leftarrow \emptyset$
 2: **for** $i = 1$ to $n$ **do**
 3:     Sample $\zeta \sim \text{Normal}(0, 1)$
 4:     $\zeta \leftarrow \zeta^{(1/d)} \times r$                                                                {Get scaling constant}
 5:     Sample $z \sim \text{Normal}(0, \mathrm{I}_d)$
 6:     $z_i \leftarrow z/|z|$                                                                                    {Normalize}
 7:     $z_i \leftarrow z_i \times \zeta$                                                                          {Project}
 8:     $y_i \leftarrow 0$
 9:     **for** $j = 1$ to $k$ **do**
10:         $y_i \leftarrow y_i + \tanh\left(z_i^j\right)$
11:     **end for**
12:     $y_i \leftarrow y_i \times (\bar{R}/k)$
13:     Add $(z_i, y_i)$ to $\mathcal{D}$
14: **end for**
15: Split $\mathcal{D}$ into $\mathcal{D}_{\text{train}}$ (80%) and $\mathcal{D}_{\text{test}}$ (20%)
16: **return** $\mathcal{D}_{\text{train}}, \mathcal{D}_{\text{test}}$

---

**Algorithm 3** Training and merging MFNNs

---

**Require:** $\mathcal{D}_{\text{train}}, \mathcal{D}_{\text{test}} = (z_{\text{test}}, y_{\text{test}}), N_{\infty}, N_{\text{list}}, M_{\max}$
**Ensure:** Dictionary *sup_norm_dic* maps $N$ to the average sup_norm
    $h_{\infty} \leftarrow$ Train a MFNN with $N_{\infty}$ neurons on $\mathcal{D}_{\text{train}}$
    $\hat{y}_{\infty} \leftarrow$ Use $h_{\infty}$ to predict on $\mathcal{D}_{\text{test}}$
    Initialize *sup_norm_dic* $\leftarrow \{\}$
    **for** $N \in N_{\text{list}}$ **do**
        $\{h_N^1, h_N^2, \ldots h_N^{M_{\max}}\} \leftarrow$ Train $M_{\max}$ MFNNs with $N$ neurons on $\mathcal{D}_{\text{train}}$
        Initialize *sup_norm_lst* $\leftarrow []$
        **for** $M \in \{1, 2, \ldots M_{\max}\}$ **do**
            *sup_norm_total* $\leftarrow 0$
            **for** 50 iterations **do**
                Randomly sample $M$ networks from $\left\{h_N^1, h_N^2, \ldots, h_N^{M_{\max}}\right\}$
                $h_{MN} \leftarrow$ Merge the $M$ networks to form a new neural network
                $\hat{y} \leftarrow$ Use $h_{MN}$ to predict on $\mathcal{D}_{\text{test}}$
                *sup_norm* $\leftarrow \max\left(|\hat{y} - \hat{y}_{\infty}|\right)$
                *sup_norm_total* $\leftarrow$ *sup_norm_total* + *sup_norm*
            **end for**
            Append *sup_norm_total*/ 50 to *sup_norm_lst*
        **end for**
        *sup_norm_dic*[$N$] $\leftarrow$ *sup_norm_lst*
    **end for**
    **return** *sup_norm_dic*

---

## B.2   Additional MFNN experiments

Beyond examining the effect of both $M$ and $N$ on sup norm, we also compare the convergence rate of MFNNs using different $\lambda \in \{10^{-1}, 10^{-2}, 10^{-3}, 10^{-4}\}$ on the multi-index regression problem. Since the training dataset is small and we intend to investigate high $\lambda$, we have to consider the low epoch setting to prevent deterioration of generalization capabilities. We train 20 networks in parallel and average the MSE (in log-scale) at each epoch, repeating this for $N \in \{300, 400, \ldots, 800\}$. Figure 2 shows that higher $\lambda$ improves the convergence speed of particles and makes training more stable. Finally, we merge networks with the same hyperparameters for comparison across different $\lambda$. A similar trend is observed in Table 2, highlighting the efficacy of PoC-based ensembling when training for fewer epochs with a high $\lambda$.

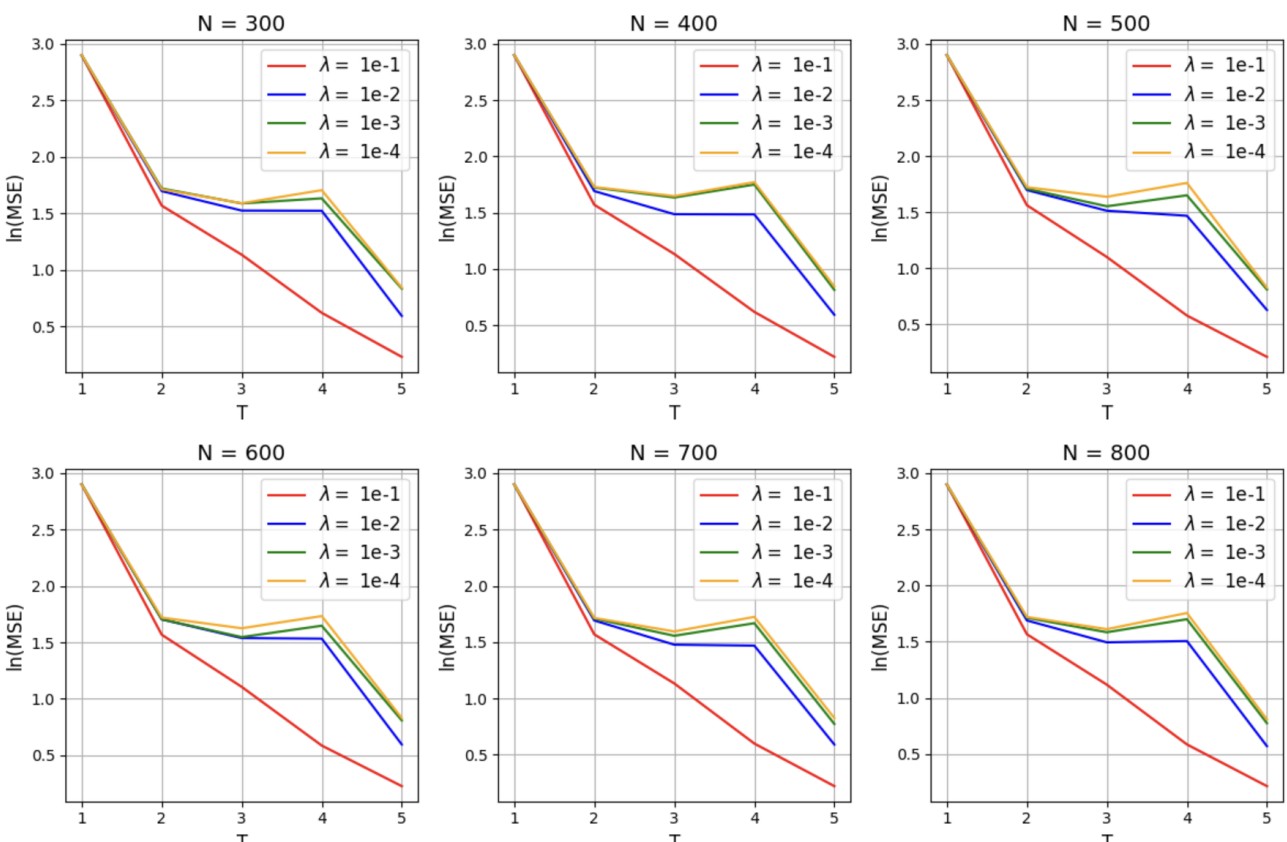

*Figure 2.* Averaged test ln(MSE) of singular MFNNs, across different $N$ and $\lambda$ for 5 epochs

*Table 2.* MSE comparison between merging $M = 20$ networks across different $N$ and $\lambda$ after 5 epochs.

| $\lambda$ | $N$ 300 | 400 | 500 | 600 | 700 | 800 |
|---|---|---|---|---|---|---|
| $10^{-1}$ | 0.9132253 | 0.9040508 | 0.9075238 | 0.9044338 | 0.9030165 | 0.9022377 |
| $10^{-2}$ | 1.2325489 | 1.2229528 | 1.2166352 | 1.1978958 | 1.1921849 | 1.1654898 |
| $10^{-3}$ | 1.5718020 | 1.5668763 | 1.5607907 | 1.5581368 | 1.5282313 | 1.5234329 |
| $10^{-4}$ | 1.6987042 | 1.6887244 | 1.6631799 | 1.6135653 | 1.5860944 | 1.5821924 |

## B.3 LoRA for finetuning language models

To examine the effect of $\lambda$, we perform LoRA and PoC-based merging by varying $\lambda \in \{0, 10^{-5}, 10^{-4}\}$ with one-epoch training. We optimize eight LoRA parameters of rank $N = 32$ in parallel using noisy AdamW with the speficied $\lambda$. Table 3 summarizes the results. For LoRA, the table lists the best result among the eight LoRA parameters based on the average accuracy across all datasets and also provides the average accuracies of the eight parameters for each dataset. We observed that for Llama2-7B with $\lambda = 0$ and $\lambda = 10^{-5}$, the chances of the optimization converging are very low. Consequently, both the average accuracy of eight LoRAs and the accuracy of PoC-based merging are also low. This is because the regularization strength $\lambda$ controls the optimization speed as seen in Theorem 3.8. On the other hand, by using a high constant $\lambda = 10^{-4}$ the average performance was improved, and PoC-based merging achieved quite high accuracy even with only one-epoch of training. This result suggests using high $\lambda$ to reduce the training costs, provided it does not negatively affect generalization error. For Llama3-8B, one-epoch training is sufficient to converge, and while LoRA performed well and PoC-based merging further improved the accuracies.

*Table 3.* Accuracy comparison of LoRA and PoC-based merging for finetuning Llama models (1 epoch).

| Model | Method | $\lambda$ | SIQA | PIQA | WinoGrande | OBQA | ARC-c | ARC-e | BoolQ | HellaSwag | Ave. |
|---|---|---|---|---|---|---|---|---|---|---|---|
| | LoRA (best) | 0 | 80.55 | 82.86 | 83.19 | 81.60 | 71.08 | 84.51 | 71.90 | 90.21 | 80.74 |
| | LoRA (ave.) | 0 | 64.73 | 76.31 | 77.76 | 68.70 | 57.02 | 69.02 | 69.04 | 70.63 | 69.15 |
| | **PoC merge** | 0 | 32.29 | 62.57 | 83.58 | 22.20 | 28.41 | 29.42 | 61.53 | 28.50 | 43.56 |
| | LoRA (best) | $10^{-5}$ | 80.14 | 82.37 | 83.43 | 80.40 | 68.86 | 83.42 | 71.68 | 89.94 | 80.03 |
| Llama2 | LoRA (ave.) | $10^{-5}$ | 74.37 | 74.12 | 80.55 | 67.50 | 58.34 | 71.98 | 69.43 | 66.25 | 70.32 |
| 7B | **PoC merge** | $10^{-5}$ | 74.56 | 83.84 | 85.16 | 60.00 | 63.14 | 78.37 | 68.72 | 92.77 | 75.82 |
| | LoRA (best) | $10^{-4}$ | 78.20 | 80.90 | 81.22 | 78.40 | 65.19 | 79.00 | 69.97 | 86.50 | 77.42 |
| | LoRA (ave.) | $10^{-4}$ | 74.42 | 77.70 | 76.08 | 75.93 | 60.93 | 76.25 | 65.68 | 66.71 | 71.71 |
| | **PoC merge** | $10^{-4}$ | 80.76 | 82.15 | 84.85 | 84.80 | 71.25 | 85.35 | 72.26 | 91.65 | 81.63 |
| | LoRA (best) | 0 | 80.45 | 88.47 | 86.82 | 87.60 | 82.25 | 90.87 | 73.85 | 95.78 | 85.76 |
| | LoRA (ave.) | 0 | 80.51 | 88.87 | 86.85 | 87.00 | 80.78 | 90.98 | 73.71 | 95.84 | 85.57 |
| | **PoC merge** | 0 | 81.73 | 88.96 | 87.77 | 88.00 | 81.40 | 91.71 | 74.46 | 96.45 | 86.31 |
| | LoRA (best) | $10^{-5}$ | 80.50 | 88.68 | 86.98 | 86.80 | 81.48 | 91.12 | 75.14 | 95.97 | 85.83 |
| Llama3 | LoRA (ave.) | $10^{-5}$ | 80.83 | 88.64 | 86.85 | 87.05 | 80.39 | 90.76 | 71.54 | 95.87 | 85.24 |
| 8B | **PoC merge** | $10^{-5}$ | 81.53 | 89.45 | 87.92 | 87.80 | 82.25 | 91.79 | 75.54 | 96.44 | 86.59 |
| | LoRA (best) | $10^{-4}$ | 80.30 | 88.57 | 86.42 | 87.20 | 78.07 | 89.81 | 73.61 | 95.14 | 84.89 |
| | LoRA (ave.) | $10^{-4}$ | 80.00 | 88.20 | 85.69 | 86.23 | 78.86 | 89.48 | 73.08 | 95.05 | 84.57 |
| | **PoC merge** | $10^{-4}$ | 80.71 | 89.72 | 88.08 | 89.00 | 82.17 | 91.79 | 74.56 | 96.36 | 86.55 |

# C  Additional Background Information

This section provides supplementary information about past works that are relevant to the paper. While not essential to the primary narrative, it will provide readers with a deeper understanding of previously established MFNN concepts and motivations behind our PoC-based model ensemble strategy.

## C.1  Mean field optimization

Two layer mean-field neural networks provide a tractable analytical framework for studying infinitely wide neural networks. As $N \to \infty$, the optimization dynamics is captured by a partial differential equation (PDE) of the parameter distribution, where convexity can be exploited to show convergence to the global optimal solution (Chizat & Bach, 2018; Mei et al., 2018; Rotskoff et al., 2019). If Gaussian noise is added to the gradient, we get MFLD which achieves global convergence to the optimal solution (Mei et al., 2018; Hu et al., 2019). Under the uniform LSI, Nitanda et al. (2022); Chizat (2022) show that MFLD converges at an exponential rate by using the proximal Gibbs distribution associated with the dynamics. MFLD has attracted significant attention because of its feature learning capabilities (Suzuki et al., 2023b; Mousavi-Hosseini et al., 2024).

As the assumption that $N = \infty$ is not applicable to real-world scenarios, a discrete-time finite particle system would align closer to an implementable MFLD i.e. noisy gradient descent. Nitanda et al. (2022) provides a global convergence rate analysis for the discrete-time update by extending the one-step interpolation argument for Langevin dynamics (Vempala & Wibisono, 2019). Meanwhile, the approximation error of the finite particle setting is studied in propagation of chaos literature (Sznitman, 1991). For finite MFLD setting, Mei et al. (2018) first suggested that approximation error grows exponentially with time before Chen et al. (2022); Suzuki et al. (2023a) proved the uniform-in-time propagation of chaos with error bound: $O\left(\frac{\alpha}{\lambda N}\right)$, suggesting that the difference between the finite $N$-particle system and mean-field limit shrinks as $N \to \infty$. However, this also means that particle approximation error blows-up exponentially as $\lambda \to 0$ due to the exponential relationship between $\alpha$ and $\lambda$ (Nitanda et al., 2022; Chizat, 2022; Suzuki et al., 2023a). Suzuki et al. (2023b) proposes an annealing procedure for classification tasks to remove this exponential dependence in LSI, wihch requires that $\lambda$ be gradually reduced over time and will not work for fixed regularization parameters. Nitanda (2024); Chewi et al. (2024) then proved a refined propagation of chaos independent of $\alpha$ at the solution as described in Section 1.1.

## C.2  Ensembling and model merging

In recent years, efforts to improve predictive capabilities and computational efficiency in machine learning have revived interest in techniques such as ensembling (Ganaie et al., 2022; Mohammed & Mohammed, 2023) and model merging (Yang et al., 2024; Goddard et al., 2024). Ensemble methods improve predictive performance by combining the outputs of multiple models during inference (Hansen & Salamon, 1990; Dietterich, 2000). Although several fusion variants exist (Kim et al., 2003; Soares et al., 2004), Cheng et al. (2018) shows that simple average voting (Breiman, 1996) does not perform significantly worse while still being highly efficient, with uses in several deep learning applications (Cheng et al., 2018; Vazquez-Romero & Gallardo-Antolin, 2020).

In contrast, model merging consolidates multiple models into a single one by combining individual parameters, showing success particularly in the optimization of LLMs (Ilharco et al., 2022; Jin et al., 2023; Davari & Belilovsky, 2024). An approach to merging models is to simply average the weights across multiple models (Utans, 1996). Taking the average of weights along a single optimization trajectory has been demonstrated to achieve better generalization (Izmailov et al., 2018; Frankle et al., 2020). Moreover, interpolating any two random weights from models that lie in the same loss basins could produce even more optimal solutions that are closer to the centre of the basin (Neyshabur et al., 2020). These works then form the foundation of *model soups* in Wortsman et al. (2022) which refers to averaging the weights of independently fine-tuned models. Similarly, Gauthier-Caron et al. (2024) showed that basic weight averaging methods can perform competitively if constituent models are similar, despite the emergence of novel LLM merging strategies (Ramé et al., 2023; Chronopoulou et al., 2023; Yu et al., 2024).

Despite the widespread use of model merging in the current research landscape, theoretical results concerning the merging of fully trained neural networks are limited. For models trained with stochastic gradient descent, averaging the model weights during different iterations of a single run improves stability bounds (Hardt et al., 2016) and variance (Jain et al., 2018) under convex assumptions. Stability bounds in the non-convex settings are then addressed by Wang et al. (2024). Ortiz-Jimenez et al. (2023) studied weight interpolation techniques for task arithmetic in vision-language models, demonstrating that

linearized models under the neural tangent kernel regime can outperform non-linear counterparts.

