# OpenReview forum: "Propagation of Chaos for Mean-Field Langevin Dynamics and its Application to Model Ensemble"
_ICML.cc/2025/Conference — ICML 2025 poster_

### Official Review · Reviewer_i5RL · 2025-03-09

**Overall Recommendation:** 3

**Summary:**

The paper studies the propagation of chaos of two-layer neural network in the mean-field regime. The authors first obtain a uniform-in-time propagation of chaos (PoC) bounds that does not depend on the LSI constant, and maintain the "original" rate of convergence.  Then, the authors apply the PoC bounds to model ensemble problem and show that  in the mean-field regime, model ensemble help reducing the approximation error. Finally, the authors proposed a PoC based model ensemble methods, and did experiments to verify the usefulness of the methods.

**Claims And Evidence:**

I think the main claims clearly stated, and the proof are convincing.

**Essential References Not Discussed:**

I don't find any important references that is not discussed.

**Experimental Designs Or Analyses:**

This paper is majorly a theoretical paper, and I think both experiments solid in experimental designs or analyses.

**Methods And Evaluation Criteria:**

This paper is majorly a theoretical paper. There are some experimental results which I think are not the major contribution of the paper: in section 5.1,  the authors consider training a two-layer neural network in mean-field regime on synthetic datasets. This experiments make sense to me as a sanity check for the theoretical results. In section 5.2, the authors consider incorporating LoRA finetuning with the proposed model ensembling method, I think the model used and the benchmarks are standard.

**Other Comments Or Suggestions:**

I do not have other comments or suggestions.

**Other Strengths And Weaknesses:**

**Strengths**
 I think in general the theoretical results are interesting:
- Removing the LSI dependency in the PoC bound and reducing the order of $1/\lambda$ in the convergence rate is a nice improvement.
- Showing that averaging over $M$ independently trained nn reduces the approximation bound (improve the PoC bound) is interesting.


**Weaknesses**

1. The main weakness in this paper is the proposed methods in Section 5.2, there are following issues in my opinion:
- 1.1  The authors take the LoRA rank to be $N$ and claim "Therefore, we can apply PoC-based model ensemble for LoRA parameters." This is not very convincing, since in  general one need large $N$ to achieve good approximation in PoC bounds, and typically $N \gg d,k.$ However, in practical application of LoRA, people let the LoRA rank $N \ll d,k,$ thus I doubt that  the PoC results provide any insight on LoRA in general.
- 1.2 There's a mismatch between this ensemble method and the one proposed section 4. In particular, the method proposed in section 4 averages the outputs, but the one in section 5.2 averages over the LoRA weights.
- 1.3 I don't see the usefulness of  this weight averaging method from the experimental results, since the authors didn't control the variates when comparing the ensembled model and the individual model. In particular, the ensembled model increases the accuracy, but also requires $M=8$ times more computation resources comparing to only training one model, since it requires to train $M=8$ independent models.  Besides, the rank of the LoRA updates in each individual model is at most $N = 32,$ but the update of  the averaged model is $N \times M = 256.$ I think it would be more interesting to consider the performance of the model ensemble method under fixed computational resources, for example,  compare with an individual model that trained $8$ times more epochs, or compare with another individual model whose LoRA rank is $N \times M = 256.$

2. While I believe the theoretical results are techinically interesting in the field of PoC,  I don't get much insight in model ensemble from the results. Theoretically, the authors show that by training $M$ independent models and then average over the output of the model, it improves the approximation error, however, (1) this results is very specific to the two-layer setting in the mean-field regime, (2) I don't see the benefit of this methods compare to directly training a large network with $M \times N$ neurons.   Practically, I don't find the practical results very convincing as discussed in the previous point.

**Questions For Authors:**

1. Could you address technical differences between the prove of the improved PoC results  and the proof techniques in [1,2] ?

[1] Chewi, S., Nitanda, A., and Zhang, M. S. Uniform-in-n log sobolev inequality for the mean-field langevin dynamics
with convex energy. arXiv preprint arXiv:2409.10440,2024.

[2] Nitanda, A. Improved particle approximation error for mean field neural networks. In Advances in Neural Information Processing Systems 37, 2024.

**Relation To Broader Scientific Literature:**

This is paper studies the propagation of chaos for two-layer neural networks in mean-field regime. In general, propagation of chaos are widely studied in other fields such as stochastic analysis/ optimal transport, statistical physics and game thoery etc. While in other problems, theoretical formulation could be different from the one considered in this paper, similar ideas might be applicable.

**Theoretical Claims:**

I check the proof of Lemma 3.6 and Proposition 4.6 in the appendix, as well as all the proof in the main context, and I think they are all correct. I didn't check the rest proofs in detail, but they are more or less easy to see given the other theorems mentioned in the main context.

---

> ### Author Rebuttal · Authors · 2025-03-30
>
> We thank the reviewer for reading our paper.
>
> **1.2 Mismatch between the ensemble methods in Sections 4 and 5.2**
>
> First, we would like to clarify that the ensemble method used in Section 5.2 is exactly the same as the one proposed in Section 4. Specifically, the ensemble in Section 4.2 is taken over model outputs of the form $x \to  b^i\_j a^{i \top}\_j x $, which reduces to parameter averaging due to the use of a linear activation function. That is: $\frac{1}{MN}\sum\_{i,j} (b^i\_j a^{i\top}\_j x) = (\frac{1}{MN}\sum_{i,j} b^i\_j a^{i\top}\_j )x = \Delta W x$ where the left-hand side represents the ensemble of model outputs, and the right-hand side corresponds to a single model with averaged parameters.
>
> **1.1 Choice of $N$**
>
> In the LoRA setting, choosing $N=\min\\{ d, k \\}$ corresponds to full fine-tuning, with no approximation error compared to the $N=\infty$ case for fixed $d$ and $k$, thanks to the linearity of the activation function. Our goal is to close the performance gap from the full-rank case more efficiently by leveraging the ensemble technique under $N<\min\\{ d, k \\}$.
>
> **1.3 Comparison under Fixed Compute Budget**
>
> Following your suggestion, we additionally evaluated LoRA with a higher rank (256) and found that performance is inferior compared to the ensemble of 8 lower-rank (32) models. Please refer to the table below for details:
>
> | Model | Method | SIQA | PIQA | WinoGrande | OBQA | ARC-c | ARC-e | BoolQ | HellaSwag | Ave. |
> | :---: | :---: | :---: | :---: | :---: | :---: | :---: | :---: | :---: | :---: | :---: |
> | Llama2 | LoRA (r=32, best)    | 79.48 | 82.43 | 81.77 | 80.60 | 67.75 | 80.47 | 70.37 | 86.67 | 78.69 |
> |               | LoRA (r=256) | 69.95 | 69.69 | 69.61 | 61.40 | 47.44 | 61.15 | 63.73 | 47.27 | 61.28 |
> |               | PoC merge     | 81.17 | 84.60 | 85.16 | 86.60 | 72.53 | 86.62 | 72.45 | 92.79 | 82.74 |
> | Llama3 | LoRA (r=32, best)    | 81.22 | 89.50 | 86.74 | 86.00 | 79.86 | 90.53 | 72.91 | 95.34 | 85.26 |
> |               | LoRA (r=256) | 81.06 | 87.60 | 87.61 | 84.60 | 78.92 |  90.06 | 75.11 |  94.98 | 84.99 |
> |               | PoC merge     | 82.04 | 89.39 | 89.27 | 89.20 | 83.28 | 92.30 | 76.33 | 96.58 | 87.30 |
>
> These results suggest that, under a fixed compute budget $MN=256$, the ensemble method achieves nontrivial improvements over joint training for a higher-rank model.
>
> Furthermore, prior studies [1] have also observed that training higher-rank matrices does not always lead to better performance, (see Fig 4 in [1]) and [2] reported instability of LoRA with higher rank.
>
> [1] S.Y. Liu et al. DoRA: Weight-Decomposed Low-Rank Adaptation. ICML, 2024
>
> [2] D. Kalajdzievski. A Rank Stabilization Scaling Factor for Fine-Tuning with LoRA. 2023
>
> **2. Benefit of the Ensemble Method**
>
> The ensemble method helps reduce the approximation error from the mean-field limit compared to directly training a single network with $MN$ neurons. Our merge strategy provides nontrivial optimal choice of $M$. Given a fixed compute budget $K=MN$, Theorem 4.4 gives the error bound: $\frac{1}{K}+\frac{1}{\sqrt{MK}}+\frac{M}{K}$, ignoring constants for simplicity. This bound decreases with $M \in [1, (K/4)^{1/3}]$, and is minimized when $M \sim (K/4)^{1/3}$, achieving an error of: $\frac{1}{K} + \frac{C}{K^{2/3}}$ for some constant $C$.
>
> Intuition: The key in the mean-field approximation is the independence among neurons $\\{h(x_t^i,z)\\}_{i=1}^N$ since the variance of their empirical average (i.e., mean-field model) would decrease linearly with $N$ if neurons were independent. While the PoC ensures that neurons become approximately independent after convergence when $N$ is sufficiently large, the ensemble of independently trained networks introduces the independence across models, further reducing error.
>
> Additionally, previous work [3] has shown that ensembles of independently trained networks can outperform joint training in specific scenarios, although their setting differs from ours.
>
> [3] Z. Allen-Zhu and Y. Li. Towards Understanding Ensemble, Knowledge Distillation and Self-Distillation in Deep Learning. ICLR, 2023
>
> **Q. Technical Differences from Existing Studies**
>
> Our proof strategy is significantly different from those in [4,5]. Specifically, [4] establishes a uniform in $N$ log-Sobolev inequality (LSI) by constructing a Lipschitz transport map from a Gaussian to the optimal distribution $\mu_*^{(N)}$. [5] directly analyzes the variance of the mean-field model at the optimal distribution, leveraging the nonlinearity of $F_0$.
>
> In contrast, our analysis is based on the argument of conditional and marginal distribution of $\mu^{(N)}$ [6], which allows us to establish an improved bound that holds for any distribution, not just at the solution. Furthermore, our assumptions about LSI differ notably from [5]; please refer to our Assumption 3.2 compared to Assumption 2 in [5].
>
> [4] S. Chewi et al., 2024
>
> [5] A. Nitanda, NeurIPS, 2024
>
> [6] F. Chen et al., 2022

---

> > ### Comment · Reviewer_i5RL · 2025-04-02
> >
> > I believe my concerns are addresses, thus  I decide to raise my score to 3.

---

### Official Review · Reviewer_7CdW · 2025-03-15

**Overall Recommendation:** 4

**Summary:**

The paper proposes an improved bound on the convergence of neurons (of a network) under mean field Langevin dynamics to an iiid distribution. This argument is known as the propagation of chaos. The convergence of the empirical finite-N distribution to the limiting iid distribution is controlled by time and number of particles (neurons) $N$. The optimal approximation error is known to be order $1/N$. This paper further improves on this and proves exponential decay in time. The result is applied to the model ensemble where the variance due to finite approximation is further reduced.

**Claims And Evidence:**

Yes, the claims are well supported.

**Essential References Not Discussed:**

The related literature is included and compared.

**Experimental Designs Or Analyses:**

Yes. The experiments in the main (with a multi-index model and two concentric circles) make sense.

**Methods And Evaluation Criteria:**

Yes. Toy models are good for theory papers.

**Other Comments Or Suggestions:**

see above.

**Other Strengths And Weaknesses:**

The results look solid but I am not an expert on this topic to catch a mistake if there was one.

However, I did find the paper difficult to read. It would benefit from better organization along the lines:

* Defective LSI comes on pg 2. This is not a trivial lemma. Please put this in a proper formatting (Lemma  ?) and cite exactly where it appeared in Chen et al 2022.

* $\Delta_0^{(N)}$ is introduced multiple times (pg3 and pg4).

* Page 3. the paragraph starting with "Afterward, this exponential dependence....." put this technical paragraph as a remark after the main result.

* Assumption 3.2 looks like a Lemma.

**Questions For Authors:**

NA.

**Relation To Broader Scientific Literature:**

Mean-field limit dynamics of neural networks is an active area of research. Improvement in convergence speed is a very valuable technical contribution. Similarly, model ensembling shows better generalization capabilities and there is no complete theory that explains this (but I may be wrong). Though this paper does not study generalization, applying mean-field Langevin analysis to ensembling is valuable.

**Theoretical Claims:**

no. I only read theorem statements in the main.

---

> ### Author Rebuttal · Authors · 2025-03-31
>
> We thank the reviewer for reading our paper and for the positive feedback. We will revise the manuscript accordingly, following your suggestion.

---

### Official Review · Reviewer_oQCo · 2025-03-16

**Overall Recommendation:** 3

**Summary:**

This paper improves the Propagation of Chaos (PoC) error bound for Mean-Field Langevin Dynamics (MFLD) by refining the defective Log-Sobolev Inequality (LSI) and introducing the Uniform Directional LSI (UD-LSI). Additionally, it proposes a PoC-based model ensemble method, which is supported by both theoretical analysis and empirical validation.

**Claims And Evidence:**

- The theoretical results heavily rely on Assumption 3.2 (Uniform directional LSI), but its justification remains insufficient. As I understand, the authors assume that the convergence rate of each conditional distribution of a Langevin particle is uniform, which indirectly ensures the network-wide uniform convergence and influences the effectiveness of the PoC-based ensembling method. However, the paper does not provide empirical evidence to support these assumptions. I encourage the authors to include numerical validation or theoretical discussion regarding the plausibility of UD-LSI in practical neural networks.
- Additionally, I am uncertain whether the improved PoC bound is numerically validated. While the theoretical derivations are rigorous, the paper does not appear to provide direct numerical verification of the improved error bound. Empirical experiments demonstrating the practical impact of the improved bound, such as comparisons with prior PoC error bounds, would strengthen the paper’s claims.

**Essential References Not Discussed:**

- This work has already discussed the related studies.

**Experimental Designs Or Analyses:**

- The paper provides empirical validation for the proposed PoC-based ensemble method, particularly in the context of LoRA-based fine-tuning.

**Methods And Evaluation Criteria:**

- The proposed method is logically well-founded and builds on existing work in the PoC for MFLD.

**Other Comments Or Suggestions:**

No.

**Other Strengths And Weaknesses:**

No.

**Questions For Authors:**

- Please refer to the comments above.

**Relation To Broader Scientific Literature:**

- While this paper makes an interesting theoretical contribution to PoC for MFLD, its relevance to the broader deep learning community remains uncertain—which is one of my main concerns.

**Theoretical Claims:**

- This paper presents rigorous theoretical proof for its main claims.

---

> ### Author Rebuttal · Authors · 2025-03-31
>
> We thank the reviewer for reading our paper.
>
> **Uniform directional LSI (UD-LSI)**
>
> We can theoretically validate the UD-LSI in the setting of Example 3.5 by leveraging a known result (e.g., Lemma 6 in [1]): Let $\nu \propto \exp( -H-V)$, where $V,H: \mathbb{R}^d \rightarrow \mathbb{R}$, with $V$ being $\alpha$-strongly convex and $H$ being $L$-Lipschitz smooth. Then, $\nu$ satisfies the Log-Sobolev Inequality (LSI) with constant: $\alpha \exp\left( - \frac{L^2}{\alpha} - \frac{4L}{\sqrt{\alpha}}\right)$. (Note that LSI constants in [1] is defined as a reciprocal number of our constant.)
> We now apply this result to the conditional distribution $\nu_{i|-i}$ in Example 3.5:
>
> $$
> \frac{d \nu\_{i|-i}}{dx}(x|\mathbf{x}^{-i}) \propto \exp\left( -\frac{N}{\lambda n}\sum_{j=1}^n\ell(\mathbb{E}\_{X\sim\rho\_{x\cup \mathbf{x}^{-i}}}[h(X,z\_j)],y\_j) - \frac{\lambda'}{\lambda}||x^i ||\_2^2 \right).
> $$
>
> The first term in the exponent is $\frac{R’}{\lambda}$-Lipschitz smooth since its partial derivative in $x$ is bounded as follows under the setting of Example 3.5:
>
> $$
> \left|\left| \frac{N}{\lambda n} \sum\_{j=1}^n \partial\_1\ell(\mathbb{E}\_{X\sim\rho\_{x\cup \mathbf{x}^{-i}}}[h(X,z\_j)],y\_j)\frac{1}{N}\partial\_x h(x,z_j)\right|\right| \leq \frac{R'}{\lambda}.
> $$
>
> And the second term in the exponent is $\frac{2\lambda’}{\lambda}$-strongly convex. Therefore, we get LSI constant $\frac{2\lambda’}{\lambda}\exp\left( -\frac{R'^{2}}{\lambda^2}  \frac{\lambda}{2\lambda'} - \frac{4R'}{\lambda }\sqrt{\frac{\lambda}{2\lambda'}}\right)
> = \frac{2\lambda’}{\lambda}\exp\left( -\frac{R'^{2}}{2\lambda \lambda'} - \frac{4R'}{\sqrt{2\lambda\lambda'}}\right)$.
>
> While this result is briefly stated in Example 3.5, we will include the above derivation in the revised version to enhance accessibility and transparency.
>
> [1] S. Chewi et al., Uniform-in-n log Sobolev inequality for the mean-field Langevin dynamics with convex energy. 2024.
>
>
> **Experiment and comparisons with prior PoC error bounds**
>
> Compared to prior results [1], which yield a uniform-in-$N$ LSI constant: $\exp\left( - \frac{1}{\lambda’} - \frac{1}{\lambda\lambda’} - \frac{1}{\lambda^2\lambda’^3}\right)$, our bound demonstrates significantly improved dependence on $\lambda, \lambda’ \to 0$, leading to faster convergence in time.
>
> Furthermore, a major improvement over [2,3] is that our particle approximation error bound is independent of $\lambda$. In contrast, earlier works suggested an exponential dependence on $\lambda$, which was overly pessimistic. To support this, we empirically investigate the effect of $\lambda$ under varying $N$ in Appendix B.2, and we did not observe such exponential blow-up, further reinforcing the practical relevance of our theoretical improvement. This highlights the importance of our contribution in tightening the gap between theoretical bounds and empirical observations. Although exactly verifying uniform bounds through experiments remains challenging, improving these theoretical bounds is an important fundamental research topic.
>
> [2] F. Chen et al., S. Uniform-in-time propagation of chaos for mean field langevin dynamics. 2022.
>
> [3] Suzuki, T., Wu, D., and Nitanda, A. Convergence of mean-field Langevin dynamics: time-space discretization, stochastic gradient, and variance reduction. NeurIPS, 2023.

---

### Official Review · Reviewer_joZ7 · 2025-03-24

**Overall Recommendation:** 2

**Summary:**

The paper establishes improved uniform-in-time propagation of chaos bounds for MFLD by removing the exponential dependence on entropy regularization, and applies this result to propose a model ensemble strategy.

**Claims And Evidence:**

The central claim of the paper is that it establishes an improved PoC result for MFLD by eliminating the exponential dependence on the $\lambda$ in the particle approximation error. This claim is clearly stated and mathematically proved in Theorem 3.7.

The derivation is supported by technical assumptions (Assumptions 3.2–3.4) and intermediate results such as Lemma 3.6. However, these assumptions are relatively strong and not well justified in practice. For example, the directional log-Sobolev inequality and the boundedness/Lipschitz conditions on model components may not hold in typical neural network architectures (e.g., ReLU activations or unbounded weights). Example 3.5 attempts to justify the assumptions but does not explicitly verify that Assumptions 3.2–3.4 are satisfied. Instead, it introduces additional constraints that further limit practical applicability.

The paper also claims a practical contribution via a model ensemble strategy derived from the theoretical insights. However, the empirical validation is limited:
- The experiments do not show how the approximation error scales with $N$ or $\lambda$, nor do they examine behavior in the small-$\lambda$ regime where prior results are known to break down.
- The ensemble setup uses $M$ independent networks of size $N$, which increases the total parameter budget and may lead to an unfair comparison under a fixed compute or model size constraint.

**Essential References Not Discussed:**

NA

**Experimental Designs Or Analyses:**

Please check in the section of Methods And Evaluation Criteria.

**Methods And Evaluation Criteria:**

The theoretical methods used in the paper seem sound. Although I am not an expert in optimal transport and did not check the proofs in the appendices in full detail, the use of Wasserstein gradient flows and functional inequalities seems fine and consistent with prior literature.

However, the evaluation criteria in the experimental section are limited and not well aligned with the core theoretical contributions. The paper does not empirically evaluate key aspects such as:
- Whether the theoretical results still hold when Assumptions 2.1 and 3.2–3.4 are violated in practice.
- How the particle approximation error scales with $N$ (number of neurons/particles),
- How performance is affected by varying the entropy regularization parameter $\lambda$,
- Whether the improved $O(1/N)$ convergence rate in Theorem 3.7 matches empirical trends,

Moreover, the ensemble strategy is only evaluated in the context of LoRA fine-tuning and lacks benchmarks on standard deep learning tasks (e.g., training deep neural networks from scratch on CIFAR-10 or ImageNet). The choice of merging $M$ independent networks of size $N$ each is not compared to more realistic alternatives under a fixed budget constraint (e.g., $M$ networks of size $N/M$), which weakens the practical relevance of the proposed approach.

**Other Comments Or Suggestions:**

NA.

**Other Strengths And Weaknesses:**

NA.

**Questions For Authors:**

1. What is the role of the regularization term $r(x)$ and the entropy regularization term in the theoretical analysis? Do the MFLD framework and the improved PoC result still hold in the absence of $L^2$ or entropy regularization?

2. Could you provide more intuition behind Assumptions 3.2 and 3.4? Specifically, how do these assumptions contribute to the analysis, and in what types of neural network architectures or setups might they realistically hold?

3. The assumption $\sup_{x,z} |h(x,z)| \le R$ seems nontrivial. In which practical scenarios or network parameterizations does this condition hold? Can you provide concrete examples?

4. The ensemble strategy seems unclear in terms of fair comparison. You consider $M$ independent networks each with $N$ neurons, but under a fixed computational or model-size budget, this setup may be suboptimal. A more realistic comparison might involve $\sqrt{N}$ networks each with $\sqrt{N}$ neurons, totaling $N$ neurons overall. In that setting, the bound in Theorem 4.4 appears similar or potentially worse due to the additional error term. Could you clarify why the proposed ensemble strategy is justified and whether it offers a real advantage under fixed resource constraints?

5. Does your theoretical framework for MFLD imply global convergence to the minimizers of the loss functionals $F_0(\mu)$ and $F_0^{(N)}(\mu^{(N)})$, similar to what is established in NTK theory? In the NTK setting, global convergence can be shown without requiring regularization. By contrast, your analysis incorporates both entropy and $L^2$ regularization. While $F_0(\mu)$ is convex over the space of distributions, it is unclear whether this alone is sufficient to guarantee global convergence of the dynamics. Could you clarify what kind of convergence your results guarantee (e.g., global vs. local), and whether additional assumptions are necessary to establish global convergence in your setting?

**Relation To Broader Scientific Literature:**

This work removes that dependence by introducing a directional log-Sobolev inequality, but the lack of comprehensive experimental study weaken the connections to broader machine learning practice.

**Theoretical Claims:**

No

---

> ### Author Rebuttal · Authors · 2025-03-29
>
> We thank the reviewer for reading our paper.
>
> __Assumption 2.1, 3.2–3.4, Example 3.5, and Q3__
>
> Assumptions 2.1, 3.2–3.4 are all satisfied in several settings considered in the mean-field Langevin literature (e.g., [1–6]). Basically, these assumptions follow for two-layer NNs with smooth and bounded activation functions. For example, under the typical loss functions (e.g., logistic and squared losses) and L2 regularization, the two-layer neural networks with the following activation function $h(x,z)$ satisfy the assumptions [4,5]: (1) $\sigma_2(r \sigma_1(w^\top z + b ))$, (2) $\sigma_2(r) \sigma_1(w^\top z +b)$, (3) $\sigma_1(w^\top z +b)$, and (4) $\sigma_1(w_1^\top z + b_1 ) + \sigma(b_2))$, where $\sigma_i$ are bounded activation functions such as tanh and sigmoid, $x=(w_1,b_1,r)$ is the parameter of each neuron, and $z$ is an input data. The last form is also discussed in Section 4.2 of our paper. While our theory does not cover ReLU activations due to their unboundedness and non-smoothness, we note that such assumptions (bounded, smooth activations) are standard in the mean-field and Langevin literature (see [3], Limitation section).
>
> The above models also satisfy the constraints introduced in Example 3.5, and thus meet Assumptions 2.1 and 3.2–3.4. Specifically, in Example we impose: (a) $\sup_{x,z}|h(x,z)|\leq R$, (b) $\ell(a,y)$ is convex and L-Lipschitz smooth w.r.t. $a\in \mathbb{R}$, and (c) $\sup_{|a|\leq R, y \in \mathcal{Y}, x \in \mathbb{R}^d, z\in \mathcal{Z}} \|\partial_1 \ell(a,y)\partial_x h(x,z)\| \leq R’$. Typical losses such as logistic and squared loss satisfy (b). Given that $\sigma_i$ are bounded and $\ell$ is L-smooth (i.e, boundedness of partial derivative w.r.t. $a$), conditions (a) and (c) are also satisfied.
>
> We will incorporate these concrete examples to improve accessibility and clarity.
>
> [1] S. Mei et al., *PNAS*, 2018
>
> [2] A. Nitanda et al., *AISTATS*, 2022
>
> [3] L. Chizat, *TMLR*, 2022
>
> [4] F. Chen et al., 2022
>
> [5] T. Suzuki et al., *NeurIPS*, 2023
>
> [6] A. Nitanda, *NeurIPS*, 2024
>
>
> __Q1/Q2__
>
> Assumption 3.4 quantifies the nonlinearity of $F_0$ with respect to the distribution. If $F_0$ is linear, MFLD reduces to a standard Langevin dynamics over $N$ independent particles. In this case, the joint distribution $\mu_t^{(N)}$ is the product measure $\mu_t^{\otimes N}$ of each particle, implying $KL(\mu_\infty^{(N)}\|\mu_*^{\otimes N})=0$ at the optimal joint distribution $\mu_\infty^{(N)}=\mu_*^{(N)}$ attained at $t=\infty$. However, in general case of nonlinear functional, there should be additional error as evaluated in Lemma 3.6; $\frac{\lambda}{N}KL(\mu_\infty^{(N)}\|\mu_*^{\otimes N})\leq \frac{B}{N}$ at the optimal solution.
> Thus, the strength of nonlinearity $B$ controls the deviation from independence among particles.
>
> Assumption 3.2 requires that the conditional distributions $\nu_{i|-i}$ satisfy an LSI, ensuring concentration of distribution of each particle. This assumption is also satisfied under the setting in Example 3.5 (for the derivation see Lemma 6 in [7]). Here, the regularization $r(x)$ is essential to encourage such concentration, and the entropy term corresponds to the Gaussian perturbation in the method.
>
> [7] S. Chewi et al., 2024
>
> __Experiments (scalability w.r.t. $M,N,\lambda$) and Q4__
>
> Scalability with respect to $M$ and $N$ is empirically validated on two-layer NN with synthetic datasets (see Fig 1). The effect of $\lambda$ is examined in Appendix Sections B.2 and B.3.
>
> Our merged method suggests a nontrivial choice of $M$ and $N$. Given a fixed computational budget $K = MN$, Theorem 4.4 yields the bound: $\frac{1}{K}+\frac{1}{\sqrt{MK}}+\frac{M}{K}$ at the solution, ignoring irrelevant constants for simplicity. This bound is decreasing on $M \in [1, (K/4)^{1/3}]$ and hence the optimal choice is $M \sim (K/4)^{1/3}$ that achieves the minimum approximation error $\frac{1}{K} + \frac{C}{K^{2/3}}$ for some constant $C$.
>
> __Global convergence and Q1/Q5__
>
> Our MFLD theory establishes global convergence of noisy gradient descent to the global minimizer of the un-regularized objective. Specifically, when $r(x) = \lambda' ||x||^2$, the regularization (i.e., $\mathbb{E}[r] + \lambda \mathrm{Ent}$) coincides with the KL divergence from a Gaussian distribution. Hence, minimizing $\mathcal{L}(\mu)$ leads to convergence toward minimizing $F_0$, up to a $\lambda$-dependent error shrinking to $0$ as $\lambda \to 0$.
>
> Importantly, optimization in the mean-field regime is more challenging than in the NTK regime, as it involves solving a truly non-convex problem. In contrast, NTK theory effectively linearizes the model and neurons evolve near initialization. Actually, the mean-field regime is known to exhibit *feature learning* behavior [8,9], deviating from NTK-regime.
>
> [8] L. Chizat et al. On Lazy Training in Differentiable Programming. *NeurIPS*, 2019
>
> [9] G. Yang and E.J. Hu. Tensor Programs IV: Feature Learning in Infinite-Width Neural Networks. *ICML*, 2021

---

> > ### Comment · Reviewer_joZ7 · 2025-04-02
> >
> > Thank you for the rebuttal. I appreciate the clarifications.
> >
> > That said, I still have some concerns—particularly regarding the bounded activation assumption, which I don’t think is trivial. In modern practice, unbounded activations like ReLU and GELU are widely used due to their optimization benefits, while bounded activations (e.g., tanh, sigmoid) can slow down training and limit expressivity. Moreover, from a statistical viewpoint, the distinction is substantial: if x is sub-Gaussian, then $\sigma(x)^$2 becomes sub-exponential when $\sigma$ is unbounded, but remains sub-Gaussian if $\sigma$ is bounded. This significantly affects tail behavior and concentration properties.
> >
> > Additionally, the experiments still don’t address key concerns:
> > - Robustness when the assumptions (e.g., boundedness) are violated;
> > - Scaling behavior with $n$ and $\lambda$ in realistic setups;
> > - Fair comparisons for the ensemble method under fixed compute or model size constraints.
> >
> > Given these factors, I decided to maintain my score.

---

> > > ### Author Response · Authors · 2025-04-03
> > >
> > > Thank you for the additional comments.
> > >
> > > First, we note that convergence results with statistical guarantees under bounded activation functions have been studied in the literature, and our work provides certain improvements in this line of research.
> > >
> > > **Global convergence under bounded activation functions.**  We would like to clarify that convergence in our setting does not necessarily imply achieving zero training error, but rather convergence to the global minimum $F(\mu_*)$ of the objective. While boundedness constraints may limit the ability to perfectly fit the training data especially when the boundedness is tight, they do not inherently make the optimization problem more difficult.
> > >
> > > To illustrate this, consider the setting where each neuron takes the form $R h(x, z)$, as used in [10], with the $L_2$ regularizer $r(x) = \lambda' ||x||^2$, where $R$ is a hyperparameter controlling boundedness and $h$ is a bounded function. As an extreme case, suppose $R\sim 0$. Then, $F_0$ becomes nearly a quadratic function which is very easy to solve. Generally, this illustrates how L2-regularization leads to a concentration of low-loss regions, facilitating the search for the global optimum.  This view is closely aligned with the perspective in the sampling theory.
> > > Actually, our theory does not currently cover ReLU, but we emphasize that boundedness does not inherently make optimization harder as seen above.
> > >
> > > **Statistical performance.** We fully agree that boundedness plays a critical role in controlling statistical performance. In  fact the paper [10] explicitly incorporates this by carefully selecting the hyperparameter $R$ to achieve strong generalization guarantees. Our convergence result can be also applied to such settings and, as discussed in our paper, offers certain theoretical improvements over prior work.
> > >
> > > **Experiments.** Since our submission is primarily theoretical, we believe additional experiments under settings that violate our assumptions (e.g., unbounded activations) are beyond the scope of the current work. That said, we have included experiments under realistic conditions with LoRA where our method shows significant improvements in accuracy under fixed compute budgets. For more details, please see our response to Reviewer i5RL.
> > >
> > > [10] T. Suzuki et al. Feature learning via mean-field langevin dynamics: classifying sparse parities and beyond. NeurIPS, 2023.

---

### Decision · Program_Chairs · 2025-05-01

**Decision:**

Accept (poster)

**Comment:**

This paper improves the error bounds for the Propagation of Chaos in Mean-Field Langevin Dynamics by introducing a new condition called the Uniform Directional Log-Sobolev Inequality. Under this condition, the main result establishes uniform-in-time convergence, with a residual term that is independent of the LSI constant of the target distribution. Furthermore, the paper improves the convergence rates with respect to key parameters, compared to previous works that established similar uniform-in-time bounds. These theoretical findings are applied to model ensembling, showing that ensemble methods can further reduce finite-size approximation error. Although primarily theoretical, the paper includes experiments designed to validate the results and demonstrate practical relevance, including an application to LoRA fine-tuning.

Most reviewers weakly support the acceptance of the paper, and I align with the general consensus.

All reviewers agree that the paper provides an interesting improvement in uniform-in-time convergence bounds for the propagation of chaos in mean-field Langevin dynamics. However, several concerns were raised, particularly regarding the application of these results to model ensembling, a point with which I also agree. I strongly encourage the authors to carefully address the reviewers' feedback in their revision, clearly highlighting the main limitations of their theoretical results and their applicability.